# Large-scale modular and uniformly thick origami-inspired adaptable and load-carrying structures

Yi Zhu [1,2] ✉ & Evgueni T. Filipov [1,2] ✉

Existing Civil Engineering structures have limited capability to adapt their configurations for new functions, non-stationary environments, or future reuse. Although origami principles provide capabilities of dense packaging and reconfiguration, existing origami systems have not achieved deployable metre-scale structures that can support large loads. Here, we established modular and uniformly thick origami-inspired structures that can deploy into metre-scale structures, adapt into different shapes, and carry remarkably large loads. This work first derives general conditions for degree-N origami vertices to be flat foldable, developable, and uniformly thick, and uses these conditions to create the proposed origami-inspired structures. We then show that these origami-inspired structures can utilize high modularity for rapid repair and adaptability of shapes and functions; can harness multi-path folding motions to reconfigure between storage and structural states; and can exploit uniform thickness to carry large loads. We believe concepts of modular and uniformly thick origami-inspired structures will challenge traditional practice in Civil Engineering by enabling large-scale, adaptable, deployable, and load-carrying structures, and offer broader applications in aerospace systems, space habitats, robotics, and more.

Civil structures are essential components of civilizations that support modern society by providing functional space and habitable shelter. To this day, we are surrounded by buildings and infrastructure built with a traditional philosophy, where structures are built slowly, offer stationary services for 50 to 100 years, and have limited options for adaptability, deconstruction, or reuse[1,2]. We envision that future civil structures need to have the following characteristics. (1) Adaptable: these structures can adapt to multiple configurations in response to changing environments and user needs (e.g. change in building floor plans). (2) Deployable: these structures can be packaged in small volumes and be rapidly deployed at target locations. Dense packaging would allow efficient transport of these systems for reuse at other locations. (3) Load-Carrying: these systems should be stiff and strong enough to serve as buildings and infrastructure. Modern structural systems still lack most of the above capabilities.

Figure 1 summarises related structural systems by measuring their functional configurations and packing ratios. The number of achievable and functional configurations of these structures represents the ability to adapt for practical uses. The packing ratio of these systems represents the ability to stow compactly for transport and reuse. State-of-the-practice civil structures can use modular designs to speed up construction[3,4]. However, these modular structures have fixed module size due to transportation limits and lack the capability to reconfigure to other structural forms. For example, cargo container-based modular structures can produce only one usable configuration without capability to reconfigure between stowed and deployed states[3] (Fig. 1A). As such, these systems have a packaging ratio of one, which is defined as the deployed volume over the packaged volume. There are also prefabricated "kit-of-parts" structural systems that can form large structures by manually assembling small individual members on site[5].

[1]Department of Mechanical Engineering, University of Michigan, Ann Arbor 48105, USA. [2]Department of Civil and Environmental Engineering, University of Michigan, Ann Arbor 48105, USA. ✉e-mail: yizhucee@umich.edu; filipov@umich.edu

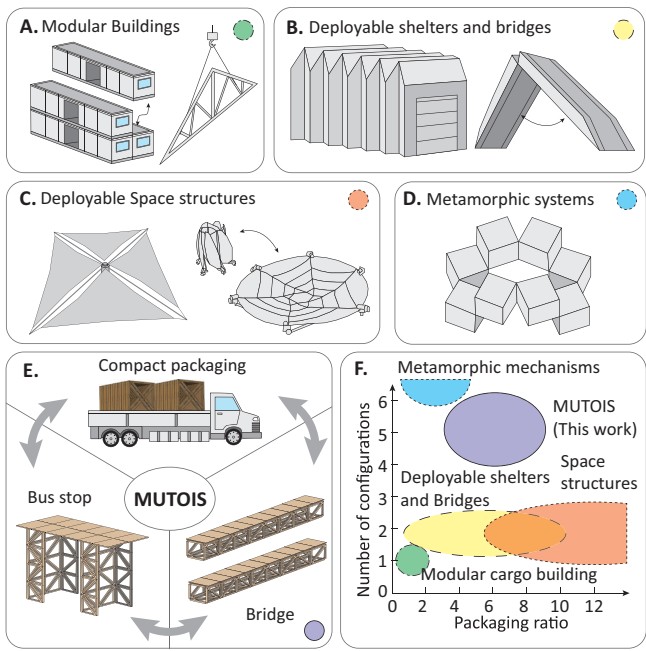

**Fig. 1 | A comparison of modular and deployable structures. A** Modular construction for buildings (cargo container modules or prefabricated members) are space inefficient in transportation and are not reconfigurable or deployable. **B** Deployable shelters and bridges offer two configurations with limited adaptability. **C** Deployable aerospace structures offer large packaging ratios but do not provide adaptable geometries. **D** Metamorphic mechanisms can achieve a large number of different configurations. However, they have limited capability to support large structural loads as civil systems. **E** This work presents Modular and Uniformly Thick Origami-Inspired Structures (MUTOIS) that offer a high packing ratio, a large load-carrying capability, and multiple configurations (see Supplementary Movie 1). **F** Comparison of different modular and/or deployable systems in terms of their ability for reconfiguration and their packaging ratios. Supplementary Note Section S1 provides details on how the representative regions are obtained.

However, structural members within these kits are not deployable and have constrained sizes to pre-specified transportation limits. Engineers have also designed deployable shelters for hazard mitigation and military operations[6] (Fig. 1B). These shelters can achieve a reasonably large packaging ratio (>5) but have a low capability to support structural loads. Figure 1B also shows deployable bridges for military operations. These systems can carry large loads at their deployed configuration and have a reasonable packaging ratio[7]. In Aerospace Engineering, deployable space structures such as solar panels, antennas, erectable booms, etc., can achieve much larger packaging ratios[8,9] (Fig. 1C). However, these systems are used in zero-gravity conditions, so their design is different from civil structures where self-weight can make significant contributions to the loading of a structure. Moreover, these above-mentioned deployable structures tend to produce only two configurations – stowed and deployed – which limits their potential for building adaptable civil structures. In Mechanical Engineering, there are metamorphic mechanisms and kinematic based metamaterial systems that can achieve a large number of different configurations (>10) through kinematic shape morphing (Fig. 1D)[10–12]. However, existing research has not shown whether these metamorphic systems have a good load-carrying capability to serve as civil structures. Furthermore, many of the configurations offered from such morphing systems do not provide useful shapes such as columns, beams, bridges, and walls that are needed for civil structures.

Therefore, this work uses origami principles to build large-scale, adaptable, reusable, and load-carrying structures to resolve the challenge (see Fig. 1E). Origami principles provide novel solutions to build engineering structures that have high packaging ratios and can change

their shapes to adapt for different functions. Origami has demonstrated applications in multiple fields including architected materials[13–15], deployable aerospace systems[16,17], robotics[18–21], and biomedical tools[22,23]. More specifically, origami and kirigami systems can utilise self-locking to form load-bearing materials[24–26] or use MDOF kinematics to achieve different configurations[10–12,27]. However, these systems target metamaterial applications or small-scale mechanisms with prototypes <1 m. Despite the tremendous progress in multiple fields, when it comes to large-scale origami for civil engineering or architectural applications, we have observed two trends: first, origami structures that can fold cannot carry large loads[28,29], and second, origami structures that can carry large loads cannot fold[30–32]. Moreover, common origami patterns for civil applications tend to have only one kinematic path[28,33], so they are not well suited for creating adaptable systems that offer multiple configurations.

In this work, we develop a Modular and Uniformly Thick Origami-Inspired Structure (MUTOIS) that can rapidly deploy from a packaged configuration, carry large loads, and achieve multiple shapes and functions for adaptability. As a highlight, Supplementary Movie 1 shows a MUTOIS prototype that has a large packaging ratio, deploys into more than five different configurations, and supports five persons as a 4-metre-long pedestrian bridge. To create these MUTOIS systems, this work first develops necessary conditions for developability and flat foldability in generic degree-N thick origami vertices (vertices with N folds). We show that among degree-4 to degree-10 vertices, there is one diamond shape degree-6 vertex that is developable, flat-foldable, uniformly thick, and has single-degree-of-freedom (SDOF) kinematics. We tessellate this diamond-shaped vertex to the Yoshimura pattern and adjust it to create the MUTOIS pattern using a generalized superimposition technique. Next, we use physical prototypes to show that high modularity in MUTOIS allows them to be repairable and adaptable. We then demonstrate a method to harness multi-path folding motions in MUTOIS for robust reconfiguration between stowed states and structural states, allowing for compact transportation and fast deployment with improved constructability and reusability. Finally, we use experiments to show that uniform thickness allows MUTOIS to have a good load-carrying capability.

## Results
### Developable, flat foldable, and uniformly thick origami
Most existing origami systems have a limited load-carrying capability because they are based on thin origami[28,29]. Although multiple thickness accommodation techniques are available in the literature[34], these techniques tend to produce non-uniform thickness where poor connectivity results in less robust load-carrying. Additionally, current thickness accommodation theories often lack thin origami benefits including rigid foldability, developability, and flat foldability. Rigid foldability allows an origami system to fold with no panel deformation, which ensures smooth folding with small actuation forces[35,36]. Developability allows an origami to rest on a flat surface when all creases are unfolded, which enables easy flat fabrication[37,38]. Flat foldability allows an origami system to fold into a compact configuration when all creases are folded[39–42]. For thin origami, prior work has introduced scalar equations, including a sector angle constraint that ensure developability[37,38], and the Kawasaki-Justin theorem and Maekawa-Justin theorem that ensure flat foldability[39–42] (see Fig. 2A "Thin Origami" column and Supplementary Table S5 for a summary). However, there exists no equivalent scalar equations for a generic degree-N thick origami vertex.

In this work, we derive necessary conditions for flat foldability and developability of an arbitrary origami vertex consisting of thick panels connected by rotational hinges (see Fig. 2B). Prior research has shown that the loop closure constraint of equivalent spatial linkages provides a necessary condition for rigid foldability in thick origami[43]. This

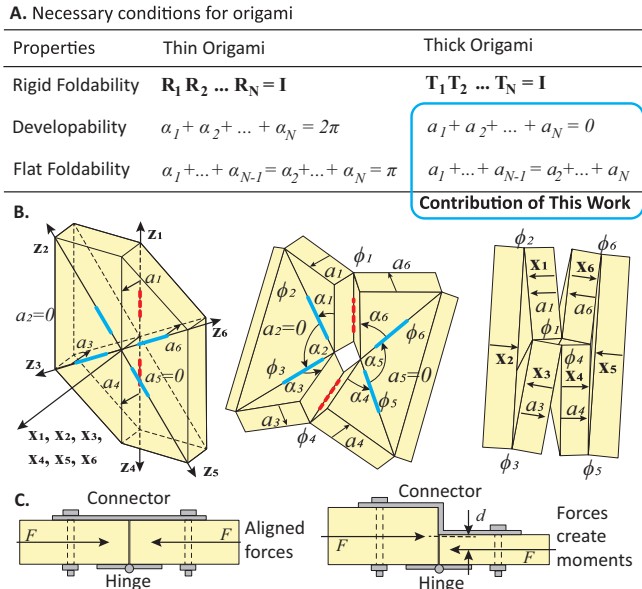

**A. Necessary conditions for origami**

| Properties | Thin Origami | Thick Origami |
|---|---|---|
| Rigid Foldability | $\mathbf{R_1 R_2 \ldots R_N = I}$ | $\mathbf{T_1 T_2 \ldots T_N = I}$ |
| Developability | $\alpha_1 + \alpha_2 + \ldots + \alpha_N = 2\pi$ | $a_1 + a_2 + \ldots + a_N = 0$ |
| Flat Foldability | $\alpha_1 + \ldots + \alpha_{N-1} = \alpha_2 + \ldots + \alpha_N = \pi$ | $a_1 + \ldots + a_{N-1} = a_2 + \ldots + a_N$ |

Contribution of This Work

**B.**

**C.**

Fig. 2 | **Necessary conditions for origami vertices. A** A summary of necessary conditions for rigid foldability, developability, and flat foldability of thin and thick origami vertices. This work derives developability and flat foldability conditions for thick origami vertices. $\mathbf{R_i}$ is rotational matrix, $\alpha_i$ is the sector angle, $\mathbf{T_i}$ is the Denavit-Hartenberg transformation matrix, $a_i$ is the thick ness offset. Detailed derivation is presented in Supplementary Note Section S2. **B** Schematic and coordinate system of a thick origami vertex with rigid foldability, developability, flat foldability, and uniform thickness. $\mathbf{x_i}$ and $\mathbf{z_i}$ are local axis, $\alpha_i$ is the sector angle, $\phi_i$ is the fold angle, $a_i$ is the thickness offset. **C** Uniform thickness enables high load-carrying capabilities after locking folding creases with gusset plates or other locking devices. With straight connector plates the connection can better transfer axial forces as well as potential bending moments.

constraint has the following form:

$$T_1 T_2 \ldots T_N = I_{4 \times 4}, \tag{1}$$

where the term $\mathbf{T_i}$ is the Denavit-Hartenberg transformation matrix expressed as:

$$
\mathbf{T_i} = \begin{bmatrix}
\cos\phi_i & -\sin\phi_i\cos\alpha_i & \sin\phi_i\sin\alpha_i & a_i\cos\phi_i \\
\sin\phi_i & \cos\phi_i\cos\alpha_i & -\cos\phi_i\sin\alpha_i & a_i\sin\phi_i \\
0 & \sin\alpha_i & \cos\alpha_i & r_i \\
0 & 0 & 0 & 1
\end{bmatrix} \tag{2}
$$

In this matrix, $\alpha_i$ are the sector angles, $\phi_i$ are the fold angles, and $a_i$ are the thickness offsets (Fig. 2B). For thick origami vertices, we have $r_i = 0$, because there is no offset in the local $\mathbf{z_i}$ axis. The demonstrated local coordinate convention in Fig. 2B is different from traditional conventions used in prior research of thick origami[43], where it is common to align the local $\mathbf{x_i}$ axis with the thickness offset $a_i$. Using this traditional convention prevents us from computing a generalized scalar equation for an arbitrary degree-N origami vertex (see Supplementary Note Section S2). Instead, the convention presented here uses the right-hand rule, defined by the counterclockwise direction of sector angles, to set up the local coordinates (as shown in Fig. 2B). This convention allows universal representation for sector angles $\alpha_i$ and fold angles $\phi_i$, and embeds a directional sign into the thickness offset $a_i$. Universal representation for sector angles $\alpha_i$ and fold angles $\phi_i$ are useful for deriving the necessary conditions for developability and flat foldability of arbitrary degree-N thick origami vertices.

For a thick origami to be developable, it needs to satisfy Eq. (1) when no creases are folded ($\phi_i = 0$). If we set $\phi_i = 0$ in Eq. (1) the first

row of the fourth column will give a thickness-based necessary condition for developability:

$$\sum_{i=1}^{N} a_i = 0 \tag{3}$$

The thickness offset $a_i$ is the distance between adjacent creases when projected to the normal vector $\mathbf{x_i}$ of panel $i$ (see Fig. 2B).

To understand Eq. (3), imagine we are walking around a thick vertex (see Fig. 2B left). Each time we move from one panel to the next panel, we need to climb up a positive offset $a_i$, climb down a negative offset $a_i$, or stay at the same elevation with $a_i = 0$. Thus, after walking around the thick vertex, we should return to where we started and recover a zero in the equation (see robust derivation in Supplementary Note Section S2.1).

For a thick origami to be flat foldable, it needs to satisfy Eq. (1) when $\phi_i = \pm 180°$. If we set $\phi_i = \pm 180°$ in Eq. (1) the first row of the fourth column will give the following thickness-based necessary condition for flat foldability:

$$a_1 + a_3 + \ldots + a_{N-1} = a_2 + a_4 + \ldots + a_N \tag{4}$$

At the flat folded configuration, the local coordinate axis $\mathbf{x_i}$ of all panels are collinear with alternating directions (see Fig. 2B right). Thus, summing the thickness offsets of all odd panels will equal the sum of thickness offsets of all even panels (see derivation in Supplementary Note Section S2.2).

Origami with uniform thickness can transfer compression and tension forces without producing moments at the connection point after the crease is closed using a locking device (Fig. 2C). In contrast, axial forces within origami with non-uniform thickness will result in unwanted moments that could limit the load-carrying performance. Supplementary Note Section S9 shows an experiment to demonstrate the superior load-carrying performance of hinged origami connections with uniform thickness. In this comparative experiment, the uniformly thick hinge is five times stiffer and has twice the ultimate capacity as compared to a similar system with uneven thickness. Moreover, having uniform thickness also enables better bending capacity, because straight connection plates have better strength and stiffness when compared to zig-zag plates (Fig. 2C). For uniformly thick vertices, the thickness offset $a_i$ can be calculated based on panel thickness $t$ and mountain-valley fold assignments as:

$$
\begin{cases}
a_i = t, & \text{if } C_i = -1, C_{i+1} = 1 \\
a_i = -t, & \text{if } C_i = 1, C_{i+1} = -1 \\
a_i = 0, & \text{if } C_i = C_{i+1}
\end{cases} \tag{5}
$$

where $C_i = -1$ means the fold is a mountain fold and it folds downward with $\phi_i < 0$ (red dotted hinge in Fig. 2B), and $C_i = 1$ means the fold is a valley fold and folds upwards with $\phi_i > 0$ (blue solid hinge in Fig. 2B). Please note that Eq. (5) is only true for uniformly thick origami systems. When applied to thick origami with extrusions or cuts in the panels, we need to directly calculate $a_i$ based on actual panel designs considering the specific cuts and extrusions.

### Creating the MUTOIS design

With the Eq. (3) to Eq. (5), one can determine if a general degree-N vertex can be developable, flat foldable, and uniformly thick. Here, we study vertices with 4 to 10 creases and summarise our findings in Fig. 3A (see details in Supplementary Note Section S3). First, odd number vertices cannot be flat foldable because they cannot satisfy the Maekawa-Justin theorem for flat foldability[39–42]. Next, we can show that degree-4 and degree-8 vertices cannot be developable, flat foldable, and uniformly thick. To prove this, we enumerate all possible mountain (M) - valley (V) assignments for degree-4 and degree-8 vertices. For

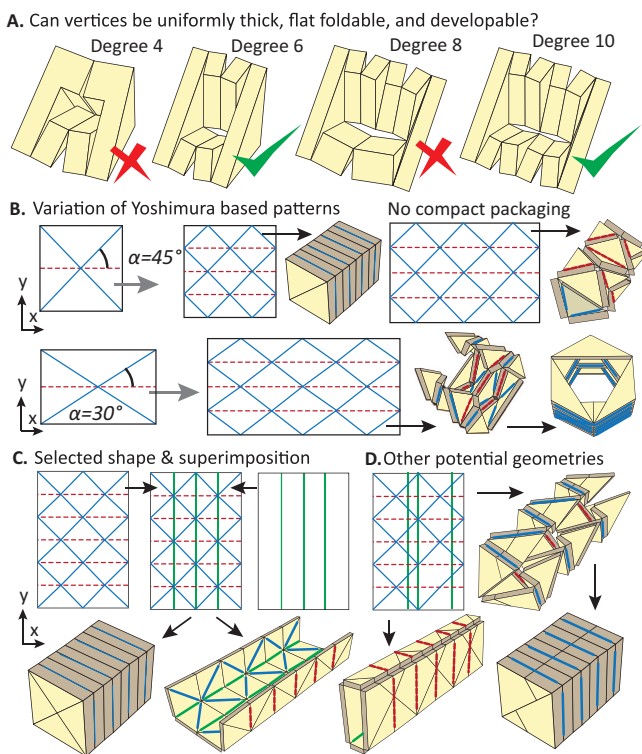

**Fig. 3 | Generating the Modular and Uniformly Thick Origami-Inspired Structure (MUTOIS) design. A** This study shows that for thick origami vertices with 4–10 creases, only one degree-6 vertex and one degree-10 vertex can be developable, flat foldable, and uniformly thick. Details are summarised in Supplementary Note Section S3. **B** Among different degree-6 vertices, only the diamond shape vertex is developable, flat foldable, and uniformly thick. This vertex can form the Yoshimura pattern. However, the basic Yoshimura pattern cannot produce long columns and has a compromised packaging ratio when the sector angle is not 45°. Red dotted lines are mountain folds while blue solid lines are valley folds. **C** Superimposition is used to construct the MUTOIS design to allow for column-like shapes of arbitrary length. Green solid lines are the superimposed folds. **D** Superimposition can build other patterns for deployable civil structures, but they no longer consist of identical triangular panels.

a degree-4 vertex, there is one MVVV assignment. For a degree-8 vertex, there are MMMMMVVV, MMMMVMVV, MMMVMMVV, MMMVMVMV, and MMVMVMMV assignments. Supplementary Note Section S3 shows that none of these mountain-valley assignments can satisfy Eq. (3) to Eq. (5). Our findings explain why known flat-panel-like 8R linkages are not based on hinged origami vertices but require cuts to form kirigami type structures[10,44].

In addition, we can show that there are degree-6 and degree-10 vertices that are developable, flat foldable, and uniformly thick. For a degree-6 vertex, there are three distinct mountain valley assignments: MMVVV, MVMVVV (waterbomb), and MVVMVV (diamond shape). Among these three configurations, only the diamond-shaped vertex can be developable, flat foldable, and uniformly thick. For a degree-10 vertex, we find that the MMVMVMMVMV assignment can produce a vertex that is developable, flat foldable, and uniformly thick.

In addition to Eq. (3)–(5), the diamond shape vertex also satisfies other conditions to ensure rigid foldability, single degree-of-freedom (SDOF) kinematics, developability, and flat foldability that were derived in previous research (summarised in Table S5). Interestingly, we find that the common Yoshimura pattern, which is based on the diamond shape vertex, can also satisfy all these conditions (Fig. 3). As such, the pattern will have smooth deployment, compact storage, and good load-carrying capability, which make it a good candidate for adaptable civil structures. The kinematic deployment of thin and thick

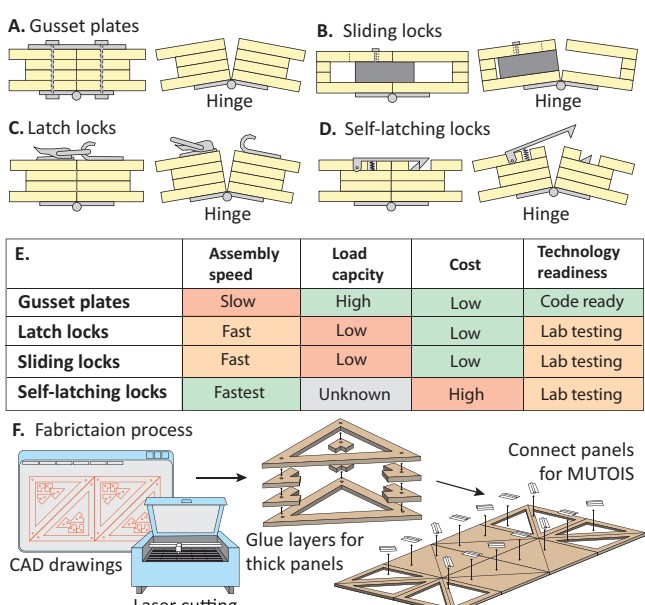

| E. | Assembly speed | Load capcity | Cost | Technology readiness |
|---|---|---|---|---|
| **Gusset plates** | Slow | High | Low | Code ready |
| **Latch locks** | Fast | Low | Low | Lab testing |
| **Sliding locks** | Fast | Low | Low | Lab testing |
| **Self-latching locks** | Fastest | Unknown | High | Lab testing |

**Fig. 4 | Locking devices and process for fabricating the Modular and Uniformly Thick Origami-Inspired Structures (MUTOIS). A** Gusset plates. **B** Sliding Locks. **C** Latch locks. **D** Self-latching locks. **E** Comparison of different locking mechanisms. Supplementary Movie 2 demonstrates how to use the different locking hinges. **F** Fabrication process of the MUTOIS.

Yoshimura patterns has been studied by previous research[45–49]. However, these works did not present generic equations on developability and flat-foldability for degree-N thick origami vertices such as Eq. (3)–(5) presented here.

Figure 3B shows variations and limitations of the basic Yoshimura pattern. First, the pattern with a 45-degree sector angle offers the most compact packaging capability, but this pattern can have at most two vertices in the x direction to achieve flat foldability without self-intersection. This means that a standard Yoshimura pattern can only build a column with an aspect ratio of four (height to width), which greatly limits its use for adaptable civil structures. To enable greater versatility, we applied a technique called superimposition of origami patterns to adjust the Yoshimura design[50]. In this work, we do not limit ourselves to SDOF origami, and instead, we superimpose patterns that produce good configurations for civil engineering structures. More specifically, we superimpose three fold lines along the y direction (Fig. 3C, D). This approach allows the design to freely expand in the y direction, producing columns and beams with an arbitrary length and different aspect ratios. Moreover, by placing the fold lines at different locations, we can produce rectangular cross-sections that are not square (see Fig. 3D). Although this design philosophy can produce a wide variety of patterns, we find that the design shown on Fig. 3C has the highest level of local modularity, with repeating identical panels that enable reusability, repairability, and adaptability. We call this design the Modular and Uniformly Thick Origami-Inspired Structures (MUTOIS).

Building upon the uniform thickness of MUTOIS, Fig. 4A–D show four different locking devices to transform MUTOIS from mechanisms to structures. Figure 4A shows a gusset plate connector, where connection plates and bolts are installed to stop folding motion in hinges. Although this connector requires the longest assembly time, it is well studied in the literature and mostly complies with current Civil Engineering codes[51]. Moreover, gusset plates have a good load-carrying capability and are well known to the current civil engineering work force. Figure 4B, C show sliding locks and latch locks for achieving more rapid assembly when compared with gusset plates. Both devices can be engaged in a matter of seconds to prevent rotation in hinges.

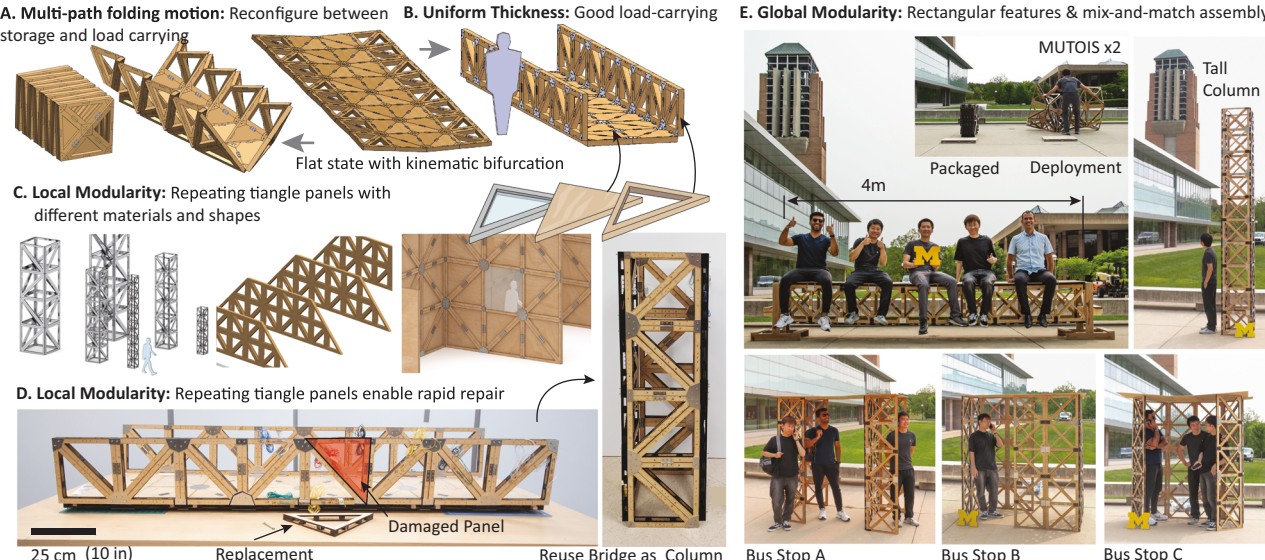

**Fig. 5 | Modular and Uniformly Thick Origami-Inspired Structures (MUTOIS) for adaptability. A** Multi-path folding motions allows MUTOIS to reconfigure between stowed and load-carrying states (Further discussion on multi-path folding is presented in Fig. 6). **B** Uniform thickness enables aligned force transfer for better load-carrying capability. Supplementary Movie 3 shows a 2-m-long physical prototype of this MUTOIS bridge that can be rapidly deployed, reconfigured, and walked on. **C** Local modularity: A MUTOIS is made up of repeating triangle panels, allowing for different materials, openings, and various structural forms. **D** Local modularity also allows MUTOIS to be rapidly repaired and reused if one or multiple panels are damaged (Supplementary Movie 4). **E** Global modularity: rectangular features allow for reconfiguration and connectivity of multiple MUTOIS units to create adaptable structures with different shapes and functions (see Supplementary Movie 1).

While these systems allow for rapid assembly, they still require manual manipulation and support smaller forces when compared with gusset plates. Finally, Fig. 4D shows a concept for a self-latching lock where assembly can be achieved without manual intervention. However, such devices require customized fabrication, higher manufacturing costs, and detailed testing to ensure sufficient strength. Figure 4E shows qualitative comparisons of the assembly speed, load carrying capacity, cost, and readiness for real world application of the four different connector designs. Supplementary Movie 2 demonstrates how to use these different locking hinges. Figure 4F shows the overall fabrication process of the proposed MUTOIS. We use a laser cutter to cut multiple layers of mid density fibreboard (MDF) for each panel, whether it is continuous or truss-like. We then connect the individual panels using aluminium hinges and locking devices. When fabricated with symmetric connection holes on both sides, the triangular panels are not subject to handedness and can be installed in any matching orientation. Further details about the fabrication and the design of MUTOIS are provided in the "Method" section and the Supplementary Note Section S4 and are supported with a laser cutter CAD file as Supplementary Data 1.

The resulting MUTOIS has three key features: (i). multi-path folding (Fig. 5A); (ii). uniform thickness (Fig. 5B); and (iii). high modularity (Fig. 5C, E). Multi-path folding motions enable MUTOIS to reconfigure into different forms and states, including a compact storage configuration for efficient transportation and reuse. For example, the MUTOIS bridge shown in Fig. 5A can fold into different kinematic paths from its flat developed state, including the storage state and the bridge state (Supplementary Movie 3). Next, having uniform thickness enables the MUTOIS to have good load-carrying capability, because force transfer between panels is aligned. The same 2-m-long bridge can support a person to walk on top (Supplementary Movie 3), while a different 4-m-long prototype can support five people (Fig. 5E and Supplementary Movie 1). Finally, MUTOIS systems have a high level of modularity which allow for versatile configurations and component designs (Fig. 5C), rapid repair (Fig. 5D), and multiple global shapes and functions for adaptable structures (Fig. 5E). For example,

Supplementary Movie 4 shows that we can rapidly repair a damaged MUTOIS bridge and repurpose it as a MUTOIS column. Supplementary Movie 1 highlights the superior adaptability of MUTOIS where we build structures with different forms and functions using two MUTOIS units and a mix-and-match assembly process.

## Local and global modularity for adaptability
Here, we show that MUTOIS offers both local modularity and global modularity that can be used to enhance the adaptability of structural systems. Local modularity is possible because MUTOIS consists of identical and repeating triangular panels. Each panel can be made with different materials (wood, metal, plastic, or encased glass), can have partial openings for function, or can have openings optimized for structural performance[52]. For example, the pedestrian bridge shown in Fig. 5A, B and Supplementary Movie 3 uses solid panels at the base for comfortable walking and truss panels on the sides for efficient load-carrying. MUTOIS systems can have an arbitrary number of units in the longitudinal direction ($y$ direction in Fig. 3C) while the number of panels in the horizontal direction ($x$ direction) can be one, two, three, or four. This flexibility in design allows MUTOIS to form a variety of customizable structures such as columns, trusses, and walls with different aspect ratios (Fig. 5C).

Figure 5D shows that the local modularity allows for rapid repair of MUTOIS systems. We had experimentally tested the MUTOIS bridge shown in Fig. 5D to failure and obtained its ultimate strength (will be discussed later). After the experiment, one triangular truss panel in the bridge was damaged. This damaged panel can be directly replaced, or remaining panels can be reused to build a different MUTOIS system. This rapid repair is possible because all triangles have the same size and geometry. When fabricated with symmetric connection holes, the triangular panels are not subject to handedness issues and can be installed in any matching orientation (see laser cutter CAD file in Supplementary Data 1). In this work, we replace the damaged truss panel and use the undamaged truss panels to build a MUTOIS column as demonstrated in Supplementary Movie 4. We show that the rebuilt MUTOIS column can maintain its foldability because the folding

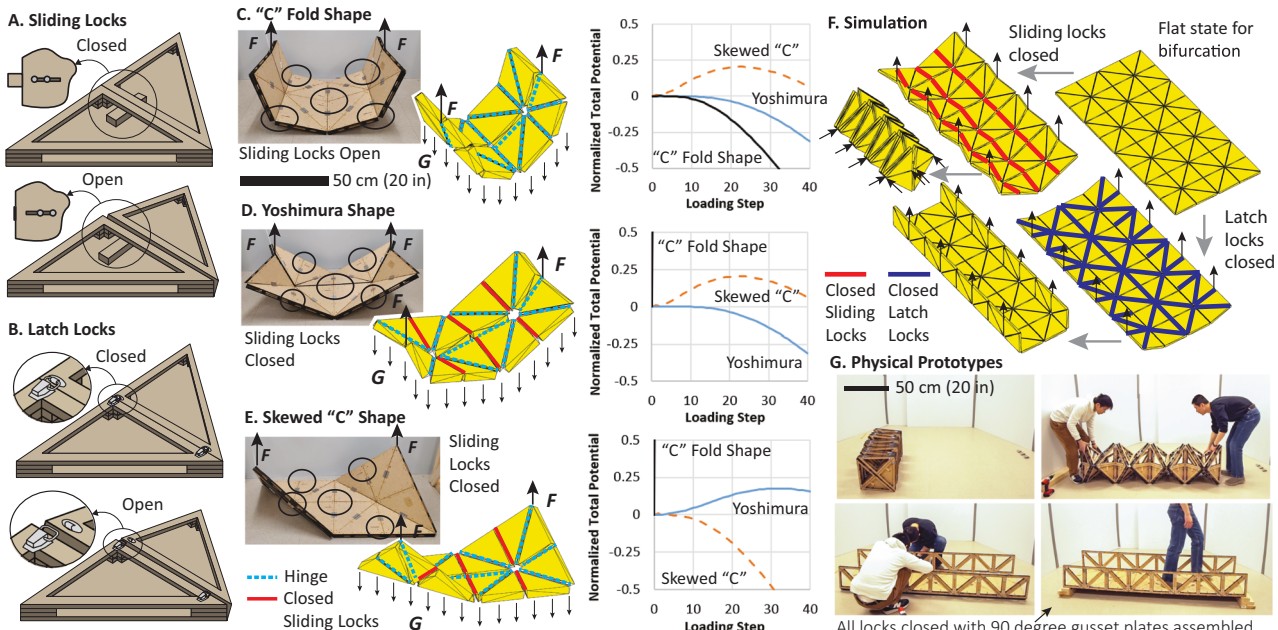

Fig. 6 | **Navigating multi-path folding in Modular and Uniformly Thick Origami-Inspired Structures (MUTOIS). A** Design of sliding locks. **B** Design of latch locks. **C–E** Simulations, experiments, and energy curves of three kinematic folding paths possible with a MUTOIS section: a "C" shape folding path (black solid lines), a Yoshimura folding path (blue solid lines), and a skewed "C" shape folding path (orange dotted lines). Supplementary Movie 5 shows comparison between simulations and experiments for **C–E. F** Bar and hinge simulations can accurately capture the folding path and the reconfiguration of a MUTOIS bridge. **G** The deployment of the physical MUTOIS bridge follows the same kinematic motions. Supplementary Movie 3 shows the deployment process of the prototype and the corresponding simulation. Source data are provided as a Source Data file.

pattern is preserved when reusing the same triangles. Moreover, this column maintains the load-carrying capability as will be shown later.

In addition, the MUTOIS has global modularity with rectangular geometric features. This allows us to connect multiple MUTOIS units to form different structural systems through a mix-and-match assembly process. Figure 5E and Supplementary Movie 1 highlight the superior adaptability of MUTOIS, where we can combine two units to build a 4-m-long bridge, a tall column, or three bus stops with different configurations. In this demonstration, we first use the compactly packaged MUTOIS sections to build the bus stop A, which takes about 40 minutes. We then reconfigure the bus stop into a 4-m-long bridge after another 45 minutes of assembly time. The 4-m-long bridge can support the weight of five persons with little deformation, highlighting its large load-carrying capability. This structure can also be used as a tall column. Next, we reconfigure the bridge to bus stops configurations B and C, which takes less than 30 min. From the bus stop configurations, reconfiguring back to the packaged state takes about 15 minutes.

The MUTOIS prototype used for the 4-m-long bridge configuration uses gusset plate connectors to obtain good load-carrying capability (shown in Fig. 4A). Although the assembly time is shorter than traditional civil construction, the gusset plates do require manual manipulation which makes the process slower and more labour intensive than that of deployable systems used in Aerospace or Mechanical Engineering. In the next section, we demonstrate how to use the latch lock connectors to improve the assembly speed, while still maintaining good load-carrying capabilities.

## Navigating multi-path folding motions

MUTOIS systems can reconfigure between stowed and structural states by harnessing multi-path folding from a flat-developed configuration. Being able to fold robustly into the stowed configuration is important for efficient transportation and reusing MUTOIS. We can navigate and enter appropriate kinematic paths by locking selected hinges and lifting the thick origami structure at specific locations. Figure 6 summarises how we can design the deployment process using the locking connectors and the proposed bar and hinge simulation methods.

Figure 6A, B show sliding locks and latch locks that we can use to stop selected creases from folding (see Supplementary Movie 2 and Supplementary Movie 5 for demonstrations). The single section MUTOIS shown in Fig. 6C–E is equipped with six sliding locks (indicated with circles in figures). If we do not lock these connectors and directly lift the MUTOIS section using the two centre nodes on both sides, the MUTOIS will naturally form a "C" shape. Even though the MUTOIS is a mechanism with MDOF and multi-path folding motions, repeating this lifting motion will consistently produce that same "C" shape, because it has the lowest total potential energy (see Supplementary Movie 5 for the experiments).

When the sliding locks are closed, the three longitudinal folding lines (red creases in Fig. 6D, E) cannot fold. In this situation, the MUTOIS section is reduced to a standard Yoshimura pattern with SDOF kinematics. The MUTOIS still has kinematic bifurcation because of the collinear creases. If we lift the structure at the middle nodes on both sides, the pattern will fold into a Yoshimura shape (Fig. 6D). If we lift the structure at the opposite corners, we obtain a skewed "C" shape (Fig. 6E).

Using kinematic analysis alone is not sufficient to identify which kinematic path the MUTOIS will fold into under a given set of boundary conditions[43,53]. However, simulating this folding behaviour as an equilibrium problem in structural analysis allows us to robustly control and design the folding process. This work uses a bar and hinge model[54,55] with formulations for thick panels and connectors to capture the behaviours accurately and efficiently (see "Method" section and Supplementary Note Section S5). The deformation path associated with the lowest total potential energy is the one that the MUTOIS will naturally enter (Fig. 6C–E). A comparison of the simulated deformations and the prototypes are provided in Supplementary Movie 5.

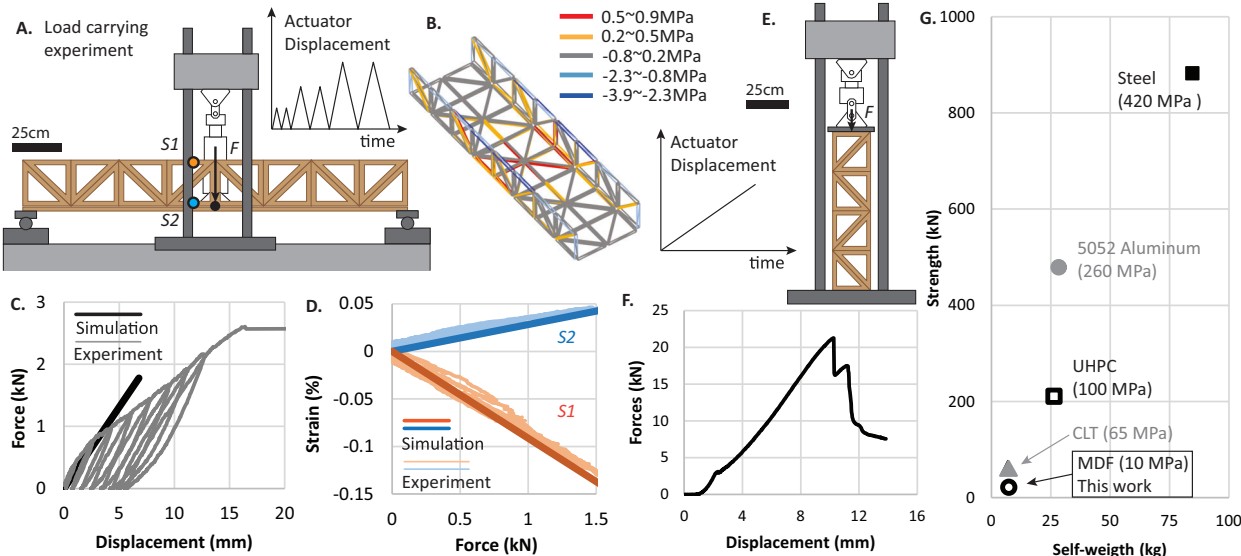

**Fig. 7 | Load-carrying capability of Modular and Uniformly Thick Origami-Inspired Structures (MUTOIS). A** Illustration of the experimental setup for the 2-m-long bridge (see experiment in Supplementary Movie 7). **B** Bar and hinge simulation for the three point bending test with computed stresses. **C** Force-displacement curve for the experiment and simulation. The 18 kg bridge can support a maximum load of 2.5 kN (250 kg). **D** Comparison of experimental and simulation data for the strain at points S1 (orange lines) and S2 (blue lines). **E** Experimental setup for the 1-m-tall MUTOIS column (see experiment in Supplementary Movie 8). **F** Force-displacement relationship, where the 7.4 kg column can support a maximum load of 21 kN (2.1 tons). **G** Strength versus weight of the MUTOIS column when scaled to other Civil Engineering materials. Source data are provided as a Source Data file.

Figure 6F, G show that the bar and hinge simulation can also capture the folding behaviours of the entire MUTOIS bridge. Before deploying the bridge, we use the simulation to verify where to lock creases and where to apply forces such that we obtain a desired deployment process. Following the simulated locking pattern and force application, we can fold the MUTOIS bridge prototype appropriately (Supplementary Movie 3). With the simulation capability, future research can explore the effects of locking a subset of creases instead of all designated creases shown in Fig. 6F. Such an investigation can help us understand whether individual creases are more critical for the transition between different kinematic paths.

The design of the individual connectors can significantly affect the assembly speed of the entire MUTOIS. In Supplementary Movie 3, the MUTOIS bridge uses 16 sliding locks, 78 latch locks, and 8 gusset plates for assembly. With two people, this bridge is assembled in 15 minutes, or a total of 30 worker minutes. In Supplementary Movie 6, we provide a comparison where the same MUTOIS bridge is assembled using 16 sliding locks and 100 gusset plates. In this scenario, the assembly process requires a total of 120 worker minutes. This result indicates that using rapid locking hinges (such as latch locks, sliding locks, or self-latching locks) can significantly improve the assembly speed and potentially lower the labour cost. A discussion on the construction of these two MUTOIS bridges can be found in Supplementary Note Section S6.

Using locking hinges and lifting the structure at selected locations is suitable for civil applications because it avoids the use of special actuators, which increase the construction cost. MUTOIS systems built with actual structural materials like steel would be much heavier than our prototypes. In such a real-world scenario, the same process would still apply - selected creases would be locked and a crane would lift the MUTOIS at appropriate locations to reconfigure it into desired shapes.

## High load-carrying capability

This section shows that thick origami can achieve much higher strength and stiffness when compared to their thin-origami counterparts[28]. We study the load-carrying capability of MUTOIS systems by experimentally testing the strength and stiffness of a 2-m-long

MUTOIS bridge and a 1-m-tall MUTOIS column. Figure 7A shows the experimental setup for the 2-m-long MUTOIS bridge where the bridge is simply supported at the two ends. We applied a displacement-controlled cyclic loading scheme to test the MUTOIS bridge where the actuator applies incremental deformations at the mid span.

Figure 7B shows a simulation of stress distributions for this loading scenario, and Fig. 7C shows the measured force-displacement result. The bridge can support 2.5 kN (250 kg) of force before failure – a considerably larger load than its self-weight of 18 kg. Using the simulated relationship between the maximum stress and the applied load (compressive $\sigma_{max} = 4.8 MPa$ at 1.5 kN load) and the compressive strength of MDF (10 MPa), we can analytically estimate the ultimate failure load of the bridge to be 3.13 kN (based on material failure). The bridge fails at 2.5 kN because buckling of the compressive truss introduces higher stress in the member than the theoretical calculation. Still, the bridge can achieve 80% of its ultimate capacity before the instability failure. At the failure force, we recorded a mid-span displacement of 15 mm, which is about 1/125 of the total bridge span and is comparable to real civil engineering structures. For example, in a real-scale testing of a high-way bridge, a mid-span displacement ratio of 1/116 was recorded at failure[56]. The initial stiffness of this bridge is 280 kN/m, which is much larger than thin-origami systems built at this scale[28]. Fig. 7D shows a good agreement between experimentally recorded strain data and simulation results. Our simulations can accurately capture the stress and strain distributions and the initial stiffness of the bridge (Fig. 7C and D); however, they cannot predict the full hysteretic behaviour. The hysteresis in the force-displacement curve originates from slacking in hinges and sliding in gusset plate connectors. The slacking and sliding are verified with optical displacement measurements of the MUTOIS bridge, where deformation within individual panels is linear elastic, while separation between panels is hysteretic (see Supplementary Note Section S7).

Figure 7E shows the load-carrying experiment setup of the 1-m-tall MUTOIS column. The column is first placed between two distribution plates and then put under the actuator, where the column is directly loaded to failure. The recorded force-displacement curve in Fig. 7F shows that the column can carry 21 kN (2.1 tons) of force, at which

point a single truss element fails. We can show that the theoretical ultimate force capacity of this column is 20.5 kN using the compressive strength of MDF, which is 10 MPa (see Supplementary Note Section S8 and S9). We can further confirm that material failure is the leading cause of failure with the recorded video of the experiment, where we found that the onset of failure is marked by delamination of MDF trusses (see Supplementary Note Section S8 and Supplementary Movie 8). This load-carrying capacity is impressive considering that the column has a self-weight of 7.4 kg and can support the weight of a large sports-utility-vehicle. This column has an initial stiffness of 1875 kN/m. At the peak force, the column experiences 10 mm of displacement at the top, which is reasonably stiff for a column built with soft MDF material and a small cross-section area of just 20.5 cm$^2$. This 1-m-tall MUTOIS column is reassembled using the undamaged panels from the 2-m-long MUTOIS bridge after the bridge is tested to failure (i.e. as shown in Fig. 5D and Supplementary Movie 4). As such, this experiment highlights that MUTOIS can maintain strength and stiffness after repair, reuse, and repurposing (see Supplementary Note Section S8 for details).

Finally, Fig. 7G extrapolates the strength and self-weight of this short column to common Civil Engineering materials including Steel, Aluminium, Ultra-High-Performance Concrete (UHPC), and Cross Laminated Timber (CLT) (see analyses in Supplementary Note Section S10). When performing this extrapolation, we assume that the column fails because of material failure only with no instability issues. We also assume that the structure maintains the same geometry and only the material properties (Young's modulus $E$, compressive strength $\sigma$, and density $\rho$) are changed. Our analysis in Supplementary Note Section S10 shows that after switching the material from MDF to steel and UHPC, the MUTOIS column should still fail due to material failure. However, when switching the material to CLT and Aluminium, the column may fail due to an instability. In this case the capacity can be smaller than what is shown in Fig. 7G (indicated with grey colour). Although testing MUTOIS systems built with different materials may eventually reveal other unforeseen failure modes, we believe that the MUTOIS concept can achieve comparable stiffness and strength when compared to non-deployable civil structures. By switching the material to structural steel (or other materials) and enlarging the cross-sections, MUTOIS can be made to achieve a comparable structural performance to non-deployable civil structures.

## Discussion

The development of MUTOIS introduces several key advancements for origami engineering at large-scales. First, this work derives thickness-based necessary conditions for developable, flat foldable, and uniformly thick degree-N origami vertices, and uses them to build the MUTOIS pattern. We showed that most degree-4 to degree-10 thick origami vertices do not satisfy these conditions, except for one type of degree-6 and one type of degree-10 vertex. Second, we fabricate metre-scale prototypes which we use to demonstrate the adaptability of MUTOIS considering both local and global modularity. Third, we develop a methodology to harness multi-path folding motions in MUTOIS to achieve appropriate reconfiguration between different storage and structural states. Fourth, we established a simulation method for MUTOIS that can capture the kinematic motion and structural load-carrying performance of MUTOIS. And fifth, we use experiments to show that MUTOIS can deploy into metre-scale adaptable structures and carry large loads, offering overall force-displacement behaviours comparable to civil engineering scale structures.

Despite the theoretical and experimental advancement, the current MUTOIS also have limitations. First, this work has not explored higher-order vertices or the degree-10 vertex that is developable, flat-foldable, and uniformly thick. Future work can explore the potential of these designs. Furthermore, this work has only experimented with

simple connection approaches without investigating the behaviours of specialised connectors (e.g. self-latching connectors shown in Fig. 3D). Because of the use of simple connection approaches, current MUTOIS systems still rely on manual assembly process that is slower than deployable mechanisms seen in Mechanical Engineering and Aerospace Engineering. We believe future research can investigate the performance of specialised connectors for thick origami systems and design new connectors that can improve load-carrying performance and to speed up assembly. In addition, integrating inexpensive actuators and exploring robotic assembly can greatly enhance MUTOIS. On the simulation side, the current bar and hinge model assumes linear elastic connector behaviours, which cannot capture the full hysteresis of MUTOIS. New nonlinear models for connectors can be integrated into origami simulations to capture the full hysteretic load-carrying behaviours. Future research can also study the embodied carbon and life-cycle cost of adaptable MUTOIS when compared to traditional non-reusable systems and/or other adaptable structural solutions. These studies will provide guidelines to engineers regarding how to pick a more appropriate structural system for a specific design scenario. Finally, it would be worthwhile to explore deployment and assembly of large and heavy MUTOIS using conventional cranes while taking advantage of the multi-path folding motions and processes explored here.

In summary, this work shows a MUTOIS that has modularity for adaptability, uses multi-path folding motions for deployment and reconfiguration, and has uniform thickness for large load-carrying capacity. The development of MUTOIS challenges the traditional thinking about civil structures – where construction is slow, and buildings become stationary objects without the ability for reconfiguration, deconstruction, or reuse. Instead, MUTOIS provides powerful tools to deploy and build structures that can adapt to non-stationary environments and user needs. Compared to other deployable structural systems such as tensegrity and deployable trusses, origami systems can directly deploy into rectangular surfaces, which are the most common geometry in civil structures. Furthermore, contrary to deployable trusses and tensegrity systems, thick origami designs can directly embed functional surfaces without compromising the system kinematics. We believe MUTOIS offers an alternative philosophy for large, adaptable, and load-carrying structures that can revolutionize how we conceive, design, build, operate, and decommission our built infrastructure. Moreover, the fundamental concepts presented here are broadly relevant to other deployable and adaptable structures such as those for aerospace systems, extra-terrestrial habitats, robotics, mechanical devices, and more.

## Methods

### Fabrication of MUTOIS

The MUTOIS prototypes are made from mid-density fibreboard (MDF) panels because this material has uniform and isotropic mechanical properties. The panel shapes are first drawn using a CAD software and then cut using a laser cutter (from Universal Laser System Inc.). After the MDF panels are cut out from the laser cutter, we stack four layers of MDF panels and glue them to build a thick MUTOIS panel. Finally, the MUTOIS panels are connected using rotational hinges. See further details in Supplementary Note Section S4 and the provided laser cutter file as Supplementary Data 1.

### Simulation of MUTOIS

This work develops a bar and hinge model, with a thick-panel-connector formulation, to simulate both the deployment kinematics and load-carrying capability of MUTOIS systems. Bar and hinge models use bar elements to capture stretching and shearing of panels and use rotational spring elements to capture crease folding[57,58]. Unlike existing bar and hinge models for thick origami[59,60], the thick-panel-connector formulation simulates thick

origami panels and connectors separately, so that we can appropriately represents their different stiffnesses. Thus, this formulation can simulate the load-carrying behaviour of MUTOIS appropriately as demonstrated in Fig. 6 and Fig. 7. A MATLAB Code Package SWOMPS[61] is used to implement this bar and hinge formulation. All the simulation codes for this paper are included in the Supplementary Code 1 and on Zenodo[62]. Further introduction of the simulation package can also be found on GitHub: https://github.com/zzhuyii/OrigamiSimulator. Further introduction of the simulation can be found in the Supplementary Note Section S5.

## Load-carrying experiments

Load-carrying experiments are done using Instron Beam Tester. Strain gauge data is collected using NI-DAQ system provided by the Structural Engineering Laboratory at the CEE department at the University of Michigan. The strain gauges we used have a 350 Ω resistance and a gauge factor of 2. Loading deformations of selected observation points are collected using an Optotrack system, which is a laser-based displacement measuring tool. Further details about the two load-carrying experiments can be found in Supplementary Note Section S7 and S8.

## Mid density fibreboard material testing

We use MTS 810 Material Test System to measure the stress-strain curve of MDF materials. The experiment follows requirements from the ASTM D3500-20 standard. 0.25-inch-thick dog-bone MDF samples are cut out using laser cutter and tested to failure using the MTS system. From the stress-strain curve, we can directly obtain Young's modulus and tensile strength of MDF. Further details can be found in Supplementary Note Section S9.

## Publication of identifiable images from human research participants

The authors affirm that human research participants provided informed consent for publication of the images in Figures and Supplementary Movies.

## Data availability

All data are available in the main text or the supplementary materials. Source data from simulation and experiments are provided as a Source Data file. Source data are provided with this paper.

## Code availability

The simulation code package used to obtain the simulation results presented in the manuscript is provided in the Supplementary Code 1. A permanent link for this version of the code is published on Zenodo[62]. Further details of the simulation can be found on GitHub: https://github.com/zzhuyii/OrigamiSimulator.

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

## Acknowledgements

This research was funded by the National Science Foundation (Grant Number: 1943723; E.T.F.) and the Automotive Research Center (ARC) (Cooperative Agreement W56HZV-19-2-0001; E.T.F.). The paper reflects the views and opinions of the authors, and not necessarily those of the funding entities. The authors want to thank Justin Roelofs, Ethan Kennedy, Steve Donajkowski, Jan Pantonlin, Guowei Tu, Anna Jia, Martin Zhou, Hardik Patil, and Mark Schenk for their support on experiments, video recording, and discussion.

## Author contributions

Conceptualization: Y.Z. Methodology: Y.Z. Investigation: Y.Z. Visualisation: Y.Z. Funding acquisition: E.T.F. Supervision: E.T.F. Writing – original draft: Y.Z. Writing – review & editing: Y.Z. and E.T.F.

## Competing interests
The authors declare no competing interests.
