## [Peer Review File · Nature Communications]

nature portfolio

Peer Review File

Large-scale modular and uniformly thick origami-inspired adaptable and load-carrying structuresEditorial Note: Parts of this Peer Review File have been redacted as indicated to remove third-party material where no permission to publish could be obtained.

REVIEWER COMMENTS

Reviewer #1 (Remarks to the Author):

This study tries to address the question: "How to make an origami-based deployable structure both load-carrying and shape adaptive?" In a bigger picture, the authors try to examine how origami structures can be genuinely valuable for real-world civil infrastructures. To this end, the research team developed a modular origami design with uniform thickness after a rigorous kinematics study and numerical simulation (using the bar-hinge method). And developed meter-scale prototypes to showcase their results using relevant material selections (aka plates, joints, and locks that satisfy the building code).

Overall, I found the research question proposed by the authors is timely, relevant, and has a broad impact. And the answers they provided are technically rigorous and convincing. This study can help the adoption of origami in practical, real-life scenarios; therefore, I recommend publication with the following suggestions for further improvements.

1. The kinematics analysis of uniformly thick origami in supplement materials is exciting and perhaps worth digging deeper into. Can you summarize your findings in the degree-4, 6, 8, and 10 vertices in the main text? Also, is the crease pattern design in Fig. 2 the only degree-6 vertex that meets all the criteria? If yes, can you prove it? If not, can you provide a few alternative designs?
2. Can you elaborate more on the kinematic bifurcation (Fig. 4a)? It's hard to see from this figure. Maybe you can add a figure focusing on only one module. Also, how would the kinematic bifurcation in this thick origami differ from thin origami with the same crease pattern? How about the bifurcation paths of the thick degree-10 vertex? Theoretically speaking, more bifurcation paths could improve the shape-adaptability of the origami.
3. How are the "locked creases" shown in Fig. 5 implemented in the experiment?

Reviewer #2 (Remarks to the Author):

Comments on "Modular and Uniformly Thick Origami for Large-Scale, Load-Carrying, and Adaptable Structures"

This manuscript aims to develop a modular and uniformly thick origami system that can rapidly deploy, carry large loads, and achieve multiple shapes and functions for adaptability. However, there are very limited significance on innovation and numerous mistakes on the kinematic fundamentals. The major issues include follows.

1. The structure proposed in this manuscript was based on the origami vertex of Yoshimura or diamond pattern, whose kinematics has been extensively explored. Meanwhile, its thick-panel forms have also been widely investigated, including non-uniform and uniform thickness [R1-R4].
[R1] Russo A, Barakali B, Kitsu K I, et al. Origami-inspired self-deployable reflectarray antenna. *Acta Astronautica*, 2023.
[R2] Jiang H, Liu W, Huang H, et al. Parametric design of developable structure based on Yoshimura origami pattern. *Sustainable Structures*, 2022, 2(2): 000019.
[R3] Yang J, You Z. Compactly folding rigid panels with uniform thickness through origami and Kirigami[C] International Design Engineering Technical Conferences and Computers and Information in Engineering Conference. American Society of Mechanical Engineers, 2019, 59247: V05BT07A042.
[R4] Zhang X, Chen Y. The diamond thick-panel origami and the corresponding mobile assemblies of plane-symmetric Bricard linkages. *Mechanism and Machine Theory*, 2018, 130: 585-604.
2. A technique called superimposition [R5] of origami patterns was used to enhance the Yoshimura pattern to generate modular and uniformly thick origami (MUTO). In [R5], crease superimposition

does not change the degrees of freedom of the original origami pattern. When Yoshimura pattern turns to eight-fold Waterbomb origami [R6], the degrees of freedom become 2.

[R5] Liu X, Gattas J M, Chen Y. One-DOF superimposed rigid origami with multiple states. *Scientific reports*, 2016, 6(1): 36883.

[R6] Grasinger M, Gillman A, Buskohl P R. Multistability, symmetry and geometric conservation in eightfold waterbomb origami. *Proceedings of the Royal Society A*, 2022, 478(2268): 20220270.

3. The method employed in this manuscript to achieve multi-shape reconfiguration relies on co-linear creases, necessitating four distinct locking devices. It is not an ingenious reconfiguration mechanism. Superimposing three folding-lines along the vertices of the Yoshimura pattern as shown in Figure 2C is redundant. Yoshimura patterns in zero-thickness and thick-panel forms have co-linear creases in themselves, and can be folded to form bridges or columns as shown in Figure 4. These redundant creases can significantly weaken the stiffness of the structure.

4. There are mistakes on the kinematic fundamentals. For example, it is mentioned "Directly using DH convention produces all-positive thickness offsets and complex sector angle representations that are not ideal for thick origami. Thus, we propose a new sign convention with universal sector angle representations and directional thickness offsets to address the problem." on page 3, line 41, and "With the traditional Denavit-Hartenberg convention, the xi axis are set to have the same direction with the thickness offset ai ...the positive direction for sector angles cannot follow a uniform counterclockwise or clockwise direction...this traditional Denavit-Hartenberg convention is not ideal for thick origami." (see Supplementary Text Section S1). In fact, link length, angle, and offset in the Denavit-Hartenberg convention can be negative.

5. A comprehensive survey on research background should be conducted. The authors mentioned "Moreover, most deployable structures used for constructing shelters and space systems produce only two configurations - stowed and deployed ...", and "Moreover, common origami patterns tend to have only one kinematic path, ...". There are many origami and modular origami inspired multi-shape reconfigurable structures, see some references [R7-R11].

[R7] Li Y, Zhang Q, Hong Y, et al. 3D transformable modular Kirigami based programmable metamaterials. *Advanced Functional Materials*, 2021, 31(43): 2105641.

[R8] Li Y, Yin J. Metamorphosis of three-dimensional kirigami-inspired reconfigurable and reprogrammable architected matter. *Materials Today Physics*, 2021, 21: 100511.

[R9] Yamaguchi K, Yasuda H, Tsujikawa K, et al. Graph-theoretic estimation of reconfigurability in origami-based metamaterials. *Materials & Design*, 2022, 213: 110343.

[R10] Liu W, Jiang H, Chen Y. 3D programmable metamaterials based on reconfigurable mechanism modules[J]. *Advanced Functional Materials*, 2022, 32(9): 2109865.

[R11] Wang C, Li J, Zhang D. Motion singularity analysis of the thick-panel kirigami[J]. *Mechanism and Machine Theory*, 2023, 180: 105162.

Reviewer #3 (Remarks to the Author):

This paper presents the Modular and Uniformly Thick Origami (MUTO) system for large-scale, load-carrying structures. The paper makes important contributions to origami geometry for large-scale applications. However, the authors should take into consideration the following comments, particularly related to the experimental investigation.

1. The statement that "Existing civil engineering structures cannot change their shape and adapt for new functions..." is incorrect. It ignores a long history of civil engineering structures that are modular, deployable, and rapidly erectable. The paper would benefit from a more rigorous literature review that includes both buildings/shelters and bridges."

2. The statement "Origami with uniform thickness can transfer compression and tension forces without producing moments (Fig. 2B) which in turn enables better load-carrying performance." is quite strange as the author later use origami in a bending environment (with flexural connections). Please clarify.

3. The connections (Fig. 3) are the most important components for the performance of large-scale structures. Providing information on the connection design, how it meets current codes (both bridge and building), and how it performs during the experiments would be critical for the paper.

4. In the videos, the connections require a high degree of manual labor. This is a weakness in the MUTO system as compared to the state-of-the-practice in modular and rapidly erectable structures. Please provide further discussion.

5. The authors have a section titled "High Load-Carrying Capability." How is "high" quantified and how does it compare to a similar structure that does not use origami?

6. The experimental aspect of the paper, and comparison with the modeling, requires major revision. Areas that should be addressed:

- Behavior of the MUTO system is not compared to a "control" specimen (e.g., a similar system that cannot deploy). This is needed to put the later discussion into context for the readers.

- The experimental test setup needs to be described in far greater depth (even in the supplemental material). For example, what are the boundary conditions?

- Stresses from simulation are compared to strains, and are stated as having "good agreement", but no quantitative comparison is made. What assumptions are being made in comparing stresses to strains? How close are they?

- The peak loads are not discussed within any context (see comment above as well).

- For the bridge, a mid-span displacement of 1/125 of the span is discussed. In typical bridge practice, the limit is 1/800 of the span. Please consider the displacement in context with current practice. Similar comments for the column.

- Important quantities like stiffness should be discussed.

- One of the most interesting aspects of the experimental program is the sliding of the connectors. Please add further discussion.

- "Material failures" are identified. However, it is not clear if the material strength is the issue as the failures seem to be occurring at points where there is added thickness between layers. Further investigation of this is necessary

7. The authors are making comparisons to other civil engineering materials on the basis that there was an MDF material failure (see comment above). However, these are all very different materials with very different behaviors. This comparison is not appropriate as it is currently performed. Instead, the authors should model each of these types of structures and compare performance, as well as weight.

Reviewer #4 (Remarks to the Author):

Modular and Uniformly Thick Origami for Large-Scale, Load2 Carrying, and Adaptable Structures
NCOMMS-23-39886-T

This paper present a highly novel type of deployable structure, based on the use of modular, thick-panel, and superimposed folding patterns. The authors have identified a clear problem in the scale-up of origami-inspired structures, and their proposed 'MUTO' system is original and demonstrated to provide a viable solution. The findings are of high significance for the structural origami research community.

The paper is in general well written, with the figures and supplementary videos in particular helping the reader to follow the theoretical and practical advancements of the MUTO system. The following items are raised as questions for minor clarification.

Comments/Questions:

INTRODUCTION SECTION

- Fig 1 - is the packaging ratios for the MUTO system calculated anywhere in text or in supplementary materials? To what extent is this influenced by the panel thickness parameter t .
- The 'number of configurations vs packaging ratio' is a novel performance metric, however what is meant for the 'modular civil structures' category shown in Fig 1? There are some kit-of-part type systems which would fall into this class and have $\# \text{configurations} \gg 1$, albeit they would not necessarily be deployable structures.
- pg 2, ln 26-30 low/high capacity to support structural loads, 'softer and lighter' - these terms seem a little bit vague. Are you referring to relaxed or n/a serviceability requirements for space structures?

'LOCAL AND GLOBAL MODULARITY' SECTION

- how many distinct modular panel types are used in the demonstrated prototypes? The paper mentions the 'solid' and 'triangular' panel types, but it is not clear if panel handed-ness or crease polarity makes these all the same or with a few different versions of each.

'NAVIGATING BIFURCATION' SECTION

- the modelling conducted for the bifurcation transition path assumes ALL creases of a particular type/subset are locked to support the lowest-energy transition to the desired state. Is locking all creases within a particular subset necessary, or will locking some/most of them sufficient?

Revisions and response to referees' comments on Manuscript NCOMMS-23-39886-T entitled: "Modular and Uniformly Thick Origami for Large-Scale, Adaptable, and Load-Carrying Structures"

by Y. Zhu and E.T. Filipov.

We are grateful to the reviewers for their feedback and helpful critique which have now brought about greater clarity to the paper. The reviewers' comments have been addressed in the manuscript, and the revisions are summarized below. Sections of the text that have been modified have been colored blue in both the manuscript and the supplementary text.

Reviewer #1 (Remarks to the Author):

This study tries to address the question: "How to make an origami-based deployable structure both load-carrying and shape adaptive?" In a bigger picture, the authors try to examine how origami structures can be genuinely valuable for real-world civil infrastructures. To this end, the research team developed a modular origami design with uniform thickness after a rigorous kinematics study and numerical simulation (using the bar-hinge method). And developed meter-scale prototypes to showcase their results using relevant material selections (aka plates, joints, and locks that satisfy the building code).

Overall, I found the research question proposed by the authors is timely, relevant, and has a broad impact. And the answers they provided are technically rigorous and convincing. This study can help the adoption of origami in practical, real-life scenarios; therefore, I recommend publication with the following suggestions for further improvements.

Reply: Thank you for giving a high evaluation of this work. We believe your suggestions have significantly improved the quality and clarity of this work. We have improved the paper based on your review.

1. The kinematics analysis of uniformly thick origami in supplement materials is exciting and perhaps worth digging deeper into. Can you summarize your findings in the degree-4, 6, 8, and 10 vertices in the main text? Also, is the crease pattern design in Fig. 2 the only degree-6 vertex that meets all the criteria? If yes, can you prove it? If not, can you provide a few alternative designs?

Reply: Thank you for these suggestions. We have now summarized the findings regarding the degree-4, 6, 8, and 10 vertices in Fig 3 of the main text. We have also summarized the general process of evaluating if a vertex can be flat foldable, developable, and uniformly thick in the main text. The following text is added:

“Here, we study vertices with 4 to 10 creases and summarize our findings in Fig. 3A (see details in Supplementary Text Section S3). First, odd number vertices cannot be flat foldable because they cannot satisfy the Maekawa-Justin theorem for flat foldability [36, 37, 38, 39]. Next, we can show that degree-4 and degree-8 vertices cannot be developable, flat foldable, and uniformly thick. To prove this, we enumerate all possible mountain (M) - valley (V) assignments for degree-4 and degree-8 vertices. For a degree-4 vertex, there is one unique MVVV assignment. For a degree-8 vertex, there are MMMMMVVV, MMMMVVV, MMMVMMVV, MMMVMVMV, and MMVMMMMV assignments. Supplementary Text Section S3 shows that none of these mountain-valley assignments can satisfy Eqn. (3) to Eqn. (5). Our findings explain why known flat-panel-like 8R linkages are not based on hinged origami vertices but require cuts to form kirigami type structures [42, 43].” (Page 6)

“In addition, we can show that there are degree-6 and degree-10 vertices that are developable, flat foldable, and uniformly thick. For a degree-6 vertex, there are three distinct mountain valley assignments: MMVVVV, MVMVVV (waterbomb), and MVVMVV (diamond shape). Among these three configurations, only the diamond shape vertex can be developable, flat foldable, and uniformly thick. For a degree-10 vertex, we find that the MMVMVMMVMV assignment can produce a vertex that is developable, flat foldable, and uniformly thick.” (Page 6)

The MMVVVV case of degree-6 vertex is rather special and interesting (see following figure). We can show that this case satisfies the proposed necessary conditions, but it has panel intersection. Thus, it is not physically possible. This case shows the limitation of our derivations: our equations are only necessary conditions. We have added additional discussion of the MMVVVV case in the supplementary material (see the new Fig. S6 in the supplementary text). This vertex is not incorporated in common origami tessellation, which is likely why it has remained unknown/unstudied in the past.

Regarding alternative patterns, the diamond shape vertex can satisfy the proposed necessary conditions and produce the Yoshimura pattern. The sector angles in the Yoshimura pattern could be defined to be different from the 45 degrees used in this work (e.g. see Fig. 3B). The waterbomb base cannot satisfy the proposed necessary conditions so it cannot be developable, flat foldable, and uniformly thick.

As a summary, the following parts in the manuscript and SI have been revised:

Manuscript: Text on Page 6; Fig. 3.

SI: Text on Page 17; Fig. S6.

2. Can you elaborate more on the kinematic bifurcation (Fig. 4a)? It's hard to see from this figure. Maybe you can add a figure focusing on only one module. Also, how would the kinematic bifurcation in this thick origami differ from thin origami with the same crease pattern? How about the bifurcation paths of the thick degree-10 vertex? Theoretically speaking, more bifurcation paths could improve the shape-adaptability of the origami.

Reply: Thank you for this suggestion. We have revised the text and have added two supplementary videos to explain the kinematic bifurcations: (1) The new supplementary video SV2 shows how the single origami vertex is locked and how we can alter the kinematics of this single vertex. (2) The new supplementary video SV5 explains the one-section MUTO system, showing how kinematic bifurcation can be adjusted through using the locking connectors. This new supplementary video SV5 is directly related to the old Fig. 4A (*which is now Fig. 5A*). We introduce these new videos when discussing Fig. 4 (SV2) and Fig. 6 (SV5) in the manuscript.

The motion of degree-6 thin vertex is similar to the thick vertex. However, the thin degree-6 vertex will have MDOF motions that are continuous so technically it is not a bifurcation/multifurcation. The thin vertex should be able to freely switch between different folding configurations due to the MDOF kinematics. When the thin vertex is loaded using gravity, it should display the same motion as the thick vertex.

The degree-10 vertex can generate more bifurcation because of having more colinear creases. However, it is unclear how to develop a tessellation using the degree-10 vertex, so we have not explored this potential. We believe this could be a potential future research direction and have added the following text in the discussion of the main text to bring this up.

“In addition to the diamond shape vertex for the MUTO pattern, this work has found another degree-10 vertex that is developable, flat-foldable, and uniformly thick. However, there is no common origami tessellation based on the degree-10 vertex so future work can explore the potential of this design.” (Page 15)

As a summary, the following parts in the manuscript and SI have been revised:

Manuscript: Text on Page 15; Captions for Fig. 4 and Fig. 6

SI: Supplementary Video SV2 and SV5

3. How are the "locked creases" shown in Fig. 5 implemented in the experiment?

Reply: Thank you for this question. The new supplementary video SV5 mentioned above also addresses this issue. We use the sliding locks to temporarily stop selected creases from folding so that we can control the folding motion of our MUTO system.

As a summary, the following parts in the manuscript and SI have been revised:

Manuscript: Captions for Fig. 6 (*please note that old Fig. 5 is now Fig 6*)

SI: Supplementary Video and SV5

Reviewer #2 (Remarks to the Author):

This manuscript aims to develop a modular and uniformly thick origami system that can rapidly deploy, carry large loads, and achieve multiple shapes and functions for adaptability. However, there are very limited significance on innovation and numerous mistakes on the kinematic fundamentals. The major issues include follows.

Reply: Thank you for taking your time to review our manuscript. While we appreciate the reviewer’s comments, *we strongly disagree with the claims that there is limited significance on innovation or that our work has mistakes regarding the kinematic fundamentals.*

First, regarding the significance of innovation in our work: the Modular and Uniformly Thick Origami (MUTO) can deploy into meter-scale structures, adapt into different structural forms, and carry remarkably large loads (hundreds of kilograms to tons). Our work derives **necessary conditions for developable and flat foldable thick origami (see detailed discussion below)**, explores modularity of MUTOs to achieve high adaptability, harnesses kinematic bifurcations for system reconfiguration, and uses experimental tests to demonstrate the high strength and stiffness of these structures. These characteristics were not previously demonstrated in the field of structural origami. We believe the significance is also reflected by the high evaluations from the other three reviewers.

Second, regarding the kinematic formulation, we believe the reviewer has misunderstood our main innovation and the technical details presented in our work. In revisiting the manuscript, we identified some potentially unclear discussions which could have led to misunderstanding regarding how we derive and use the new theoretical findings. We have revised these sections, and we address the specific comments brought up by the reviewer in the forthcoming responses.

More specifically, one key achievement in our kinematic derivation is to obtain **scalar equations** for necessary conditions of developability and flat foldability in a **generic degree-N vertex**. While past research has shown the sector angle constraint $\sum \alpha_i = 2\pi$ for developability and the Kawasaki-Justin theorem $\alpha_1 + \alpha_3 + \dots + \alpha_{n-1} = \alpha_2 + \alpha_4 + \dots + \alpha_N = \pi$ for flat foldability, there are no similar equations derived for thick origami.

This work showed that we can derive similar scalar equations for thick origami vertices, including $\sum_{i=1}^N a_i = 0$ for ensuring developability and $a_1 + a_3 + \dots + a_{N-1} = a_2 + a_4 + \dots + a_N$ for ensuring flat foldability in thick origami. *In these two equations, a_i is the thickness offset of panels.* These two equations have not been derived in previous research and cannot be found in the papers suggested by the reviewer. To better reflect this contribution, we have revised **Figure 2** in the manuscript to explicitly show these necessary conditions for thick origami.

A. Necessary conditions for origami		
Properties	Thin Origami	Thick Origami
Rigid Foldability	$R_1 R_2 \dots R_N = I$	$T_1 T_2 \dots T_N = I$
Developability	$\alpha_1 + \alpha_2 + \dots + \alpha_N = 2\pi$	$a_1 + a_2 + \dots + a_N = 0$
Flat Foldability	$\alpha_1 + \dots + \alpha_{N-1} = \alpha_2 + \dots + \alpha_N = \pi$	$a_1 + \dots + a_{N-1} = a_2 + \dots + a_N$
-	-	Contribution of This Work

Furthermore, our work has applied the new equations to study degree-4 to 10 vertices, showing that most thick origami vertices cannot be developable, flat foldable, and uniformly thick, except for a diamond

shape degree 6 vertex and a degree 10 vertex. This additional study is now highlighted in Figure 3 of the paper. *Again, we believe these theoretical contributions are significant and were not shown in prior research as pointed out by Reviewer 1.*

To address the comment, the following sentence is rewritten in the abstract to highlight the contribution of this work:

“This work first derives general conditions of degree-N thick origami vertices to be flat foldable and developable and uses these conditions to create the MUTO pattern.” (Page 1)

In addition, we have summarized the major development in the introduction to bring up our contribution:

“To create these MUTO systems, this work first develops necessary conditions for developability and flat foldability in generic degree-N thick origami vertices (vertices with N folds). We show that among degree-4 to degree-10 vertices, there is one diamond shape degree-6 vertex that is developable, flat-foldable, uniformly thick, and has single-degree-of-freedom (SDOF) kinematics. We tessellate this diamond shape vertex to the Yoshimura pattern and adjust it to create the MUTO pattern using a generalized superimposition technique.” (Page 3)

As a summary, the following parts in the manuscript and SI have been revised:

Manuscript: Fig. 2. Text on Page 1 and Page 3

1. The structure proposed in this manuscript was based on the origami vertex of Yoshimura or diamond pattern, whose kinematics has been extensively explored. Meanwhile, its thick-panel forms have also been widely investigated, including non-uniform and uniform thickness [R1-R4].

[R1] Russo A, Barakali B, Kitsu K I, et al. Origami-inspired self-deployable reflectarray antenna. Acta Astronautica, 2023.

[R2] Jiang H, Liu W, Huang H, et al. Parametric design of developable structure based on Yoshimura origami pattern. Sustainable Structures, 2022, 2(2): 000019.

[R3] Yang J, You Z. Compactly folding rigid panels with uniform thickness through origami and Kirigami, International Design Engineering Technical Conferences and Computers and Information in Engineering Conference. American Society of Mechanical Engineers, 2019, 59247: V05BT07A042.

[R4] Zhang X, Chen Y. The diamond thick-panel origami and the corresponding mobile assemblies of plane-symmetric Bricard linkages. Mechanism and Machine Theory, 2018, 130: 585-604.

Reply: Thank you for the comment and mentioning these related works. Yes, the final structure we designed is based on a widely studied Yoshimura pattern. However, the main contribution of this work (on kinematic analysis of origami) is on deriving necessary conditions for developability and flat foldability of generic thick origami vertices (see also the comments above). **These necessary conditions are applicable to generic degree-N vertices.** These include the following two equations:

$$\sum_{i=1}^N a_i = 0$$

$$a_1 + a_3 + \dots + a_{N-1} = a_2 + a_4 + \dots + a_N$$

Although the references [R1-R4] have studied the kinematics of thin and thick Yoshimura patterns, they only studying the specific pattern, and do not present formulations that can be generalized to other patterns. More importantly, these papers do not answer the question of: *How to find a developable, flat foldable, and uniformly thick origami vertex?*

The new theory developed by our work gives us a way of investigating thick origami of arbitrary degree-N vertices. Moreover, our work shows that among degree 4, 5, 6, 7, 8, 9, and 10 vertices, only the diamond shape vertex and one degree-10 vertex can be developable, flat-foldable, and can have uniform finite thickness. The detailed derivations are presented in Supplementary Text Section S3

Thus, we do not think the mentioned references [R1-R4] weaken the significance of this work. Instead, our paper presents a generalized method that can be used to explore the design space of thick origami, which is beyond what previous work has been able to show. We have revised the text and Figures to make these contributions clearer. The following text is added in the manuscript to address this comment:

“The kinematic deployment of thin and thick Yoshimura patterns has been studied by previous research [44, 45, 46, 47]. However, these works did not present generic equations on developability and flat-foldability for degree-N thick origami vertices such as the Eqn. (3) to Eqn. (5) presented here.” (Page 7)

As a summary, the following parts in the manuscript and SI have been revised:
 Manuscript: **Fig. 2**. Text on **page 7**.

2. A technique called superimposition [R5] of origami patterns was used to enhance the Yoshimura pattern to generate modular and uniformly thick origami (MUTO). In [R5], crease superimposition does not change the degrees of freedom of the original origami pattern. When Yoshimura pattern turns to eight-fold Waterbomb origami [R6], the degrees of freedom become 2.

[R5] Liu X, Gattas J M, Chen Y. One-DOF superimposed rigid origami with multiple states. Scientific reports, 2016, 6(1): 36883.

[R6] Grasinger M, Gillman A, Buskohl P R. Multistability, symmetry and geometric conservation in eightfold waterbomb origami. Proceedings of the Royal Society A, 2022, 478(2268): 20220270.

Reply: Thank you for raising this point. We agree that there is a subtle difference in how we use superposition in our work compared to [R5], and this difference did require some further elaboration. We use a **more generic** superimposition concept which does add degrees of freedom initially. However, we can also lock selected creases in the eight-fold system to reduce it back to a six-fold system (with a single degree of freedom). **The new Supplementary Video SV2 explains how this can be achieved.**

With the basic form of superimposition, SDOF kinematics are preserved because only SDOF patterns are superimposed on top of each other. In this case folding into one pattern restricts the motion of the other pattern. In our work, we break this SDOF constraint and allow for superimposition to form MDOF patterns initially. This more generic form of superposition is necessary to enable a broader range of geometric reconfiguration, and to allow for columns and beams with arbitrary aspect ratios (see figure above). The above figure is added to the manuscript to further explain the reconfiguration and dimensional opportunities offered by this more generic form of superposition. Also, to clarify how we use the enhanced superimposition concept, the following sentence is rewritten in the manuscript:

“To enhance the versatility, we applied a technique called superimposition of origami patterns to adjust the Yoshimura pattern [48]. In this work, we do not limit ourselves to SDOF origami patterns. Instead, we superimpose patterns that produce good civil structure configurations.” (Page 8)

Furthermore, as the reviewer pointed out, while the degree-8 vertex has 2 DOFs, this is not a problem because we can use locking connectors to change the kinematics. We have added two new **Supplementary Videos SV2** and **SV5** to explain how we can temporarily lock selected creases to control the deployment of the system. Thus when folding the MUTO system using the Yoshimura kinematic path, the eight-fold vertex is actually reduced back to a six-fold vertex by locking two creases. The locking and deployment processes are shown in detail in the new **Supplementary Video SV2** and **SV5**.

As a summary, the following parts in the manuscript and SI have been revised:

Manuscript: **Fig. 3**. Text on **Page 8**.

SI: **Supplementary video SV2** and **SV5**.

3. The method employed in this manuscript to achieve multi-shape reconfiguration relies on co-linear creases, necessitating four distinct locking devices. It is not an ingenious reconfiguration mechanism. Superimposing three folding-lines along the vertices of the Yoshimura pattern as shown in Figure 2C is redundant. Yoshimura patterns in zero-thickness and thick-panel forms have co-linear creases in themselves, and can be folded to form bridges or columns as shown in Figure 4. These redundant creases can significantly weaken the stiffness of the structure.

Reply: Thank you for raising this comment which again relates to the more generic form of superimposition. While we respect the reviewer’s comments, we kindly disagree, and again believe there is a misunderstanding regarding the underlying concept here. We address their comment in two parts:

First, without superimposition, the Yoshimura pattern *cannot* produce a column with arbitrary aspect ratios while maintaining compact packaging. In fact, as shown in the figure above for the basic Yoshimura pattern to maintain a compact packaging state, *it can only have two sections* in the x-direction.

This means that the aspect ratio of the column (height over width) can only be four. Adding more sections in the x-direction will prevent the basic Yoshimura pattern from folding into a compact packaging state. The text in the manuscript and Figure 3 are updated to better address this concept:

“Figure 3B shows variations and limitations of the basic Yoshimura pattern. First, the pattern with a 45-degree sector angle offers the most compact packaging capability. More importantly however, this pattern can have at most two vertices in the x direction to achieve flat foldability without self-intersection. This means that a standard Yoshimura pattern can only build a column with an aspect ratio of four (height to width), which greatly limits its use for adaptable civil structures.” (Page 8)

Second, while these redundant creases may reduce stiffness of the structure, we do not agree that this effect is significant, or that it is a problem for our work. When assembled using proper locking creases, our systems can make:

- (i) a 4-meter bridge that supports the weight of 5 persons with little deformation – Figure 5,
- (ii) a 2-meter C-shaped bridge that can carry 2.5 kN with a displacement of 15mm – Figure 7,
- (iii) a 1-meter-tall column that can carry 2.1 tons with a displacement of 10mm – Figure 7.

To our best knowledge the demonstrated performance is stiffer and stronger than other deployable meter-scale origami structures in the literature.

As a summary, the following parts in the manuscript and SI have been revised:
Manuscript: Fig. 3. Text on Page 8.

4. There are mistakes on the kinematic fundamentals. For example, it is mentioned “Directly using DH convention produces all-positive thickness offsets and complex sector angle representations that are not ideal for thick origami. Thus, we propose a new sign convention with universal sector angle representations and directional thickness offsets to address the problem.” on page 3, line 41, and “With the traditional Denavit-Hartenberg convention, the x_i axis are set to have the same direction with the thickness offset a_i ...the positive direction for sector angles cannot follow a uniform counterclockwise or clockwise direction...this traditional Denavit-Hartenberg convention is not ideal for thick origami.” (see Supplementary Text Section S1). In fact, link length, angle, and offset in the Denavit-Hartenberg convention can be negative.

Reply: Thank you for raising this point. While we respect the reviewer’s comments, we again believe there is a misunderstanding regarding the derivation we implemented. We have thus revised the text to improve the description and avoid misunderstandings.

“The demonstrated local coordinate convention in Fig. 2B is different from traditional conventions used in prior research of thick origami [40, 41], where it is common to align the local x_i axis with the thickness offset a_i . Using this traditional convention prevents us from computing a generalized scalar equation for an arbitrary degree-N origami vertex (see Supplementary Text S2). Instead, the convention presented here uses the right-hand rule, defined by the counterclockwise direction of sector angles, to set up the local coordinates (as shown in Fig. 2B). This convention allows universal representation for sector angles α_i and fold angles ϕ_i , and embeds a directional sign into the thickness offset a_i . Universal representation for sector angles α_i and fold angles ϕ_i are useful for deriving the necessary conditions for developability and flat foldability of arbitrary degree-N thick origami vertex.” (Page 4)

Furthermore, while we agree with the reviewer that the link length, angle, and offset within an DH matrix for a general linkage can be both positive and negative. Directly using DH convention may not be the most convenient approach for thick origami under all circumstances. In prior work where the DH matrix is applied to study thick origami, it is most common to use the convention demonstrated in (Y. Chen et. al., 2015, Origami of thick panels, *Science* 349, 396). With this convention the thickness offsets $a_{i(i+1)}$ are aligned with the local axis x_i (see the following figures). Because of this selection, **thickness offset $a_{i(i+1)}$ is positive (also for the final one a_{N1})**. This selection is commonly seen in other papers on thick origami vertices (for example [R4] mentioned by the reviewer).

[REDACTED]

Sign convention from Chen et al 2015 and [R4] mentioned by the reviewer.

This selection works well for analyzing the kinematics of a specific vertex, but it does not have universal representation for the kinematic variable θ angles and the sector angles α in the DH matrix. We need to use different equations to convert the dihedral angles to the kinematic variable θ angles. For example, in the Chen et. al. 2015 paper, different sets of equations are used for the diamond shape vertex and for the water bomb vertex:

$$\theta_1 = \pi - \varphi_1, \theta_2 = \pi + \varphi_2, \theta_3 = \pi + \varphi_3,$$

$$\theta_4 = \pi - \varphi_4, \theta_5 = \pi + \varphi_5, \theta_6 = \pi + \varphi_6, \quad (\text{Diamond shape})$$

$$\theta_1 = \pi + \varphi_1, \theta_2 = \pi + \varphi_2, \theta_3 = \pi - \varphi_3,$$

$$\theta_4 = \pi + \varphi_4, \theta_5 = \pi - \varphi_5, \theta_6 = \pi + \varphi_6. \quad (\text{Waterbomb shape})$$

As shown above, even for origami vertices with the same number of creases, using traditional DH convention will force users to apply different equations to finish the calculation. This prevents us from deriving thickness equations for developability and flat foldability in general degree-N origami vertices, where **generality** is significant. Because of the issues discussed above, our work adopts another sign convention to analyze the thick origami vertices. Details of our derivation are shown in the supplementary

text Section S2.1 and S2.2. **The main contribution of this work is highlighted in the following figure, and again we disagree that there are mistakes related to how we derive these equations.**

A. Necessary conditions for origami		
Properties	Thin Origami	Thick Origami
Rigid Foldability	$R_1 R_2 \dots R_N = I$	$T_1 T_2 \dots T_N = I$
Developability	$\alpha_1 + \alpha_2 + \dots + \alpha_N = 2\pi$	$a_1 + a_2 + \dots + a_N = 0$
Flat Foldability	$\alpha_1 + \dots + \alpha_{N-1} = \alpha_2 + \dots + \alpha_N = \pi$	$a_1 + \dots + a_{N-1} = a_2 + \dots + a_N$
		Contribution of This Work

As a summary, the following parts in the manuscript and SI have been revised:
 Manuscript: Text to **Page 4**; **Fig. 2**.

5. A comprehensive survey on research background should be conducted. The authors mentioned “Moreover, most deployable structures used for constructing shelters and space systems produce only two configurations - stowed and deployed ...”, and “Moreover, common origami patterns tend to have only one kinematic path, ...”. There are many origami and modular origami inspired multi-shape reconfigurable structures, see some references [R7-R11].

[R7] Li Y, Zhang Q, Hong Y, et al. 3D transformable modular Kirigami based programmable metamaterials. *Advanced Functional Materials*, 2021, 31(43): 2105641.

[R8] Li Y, Yin J. Metamorphosis of three-dimensional kirigami-inspired reconfigurable and reprogrammable architected matter. *Materials Today Physics*, 2021, 21: 100511.

[R9] Yamaguchi K, Yasuda H, Tsujikawa K, et al. Graph-theoretic estimation of reconfigurability in origami-based metamaterials. *Materials & Design*, 2022, 213: 110343.

[R10] Liu W, Jiang H, Chen Y. 3D programmable metamaterials based on reconfigurable mechanism modules[J]. *Advanced Functional Materials*, 2022, 32(9): 2109865.

[R11] Wang C, Li J, Zhang D. Motion singularity analysis of the thick-panel kirigami[J]. *Mechanism and Machine Theory*, 2023, 180: 105162.

Reply: Thank you for pointing out these references. They are relevant and we have included them in our paper to help readers better appreciate the contribution of this work.

Although these references do show mechanisms that can achieve multiple configurations and potentially large packaging ratio, these references are focusing on metamaterial type applications. There are two limitations that prevents us from using these systems for adaptable civil structures. First, these systems have not been shown to have the capability to support large loads. Second, despite having a large number of different configurations, not all configurations are usable. It is unclear how to design these systems to have direct application to different civil structural shapes like bridges, beams, columns, walls, etc. With this in mind, we think our proposed MUTO system still demonstrates a unique and superior

adaptability for a civil engineering context. To address the reviewer’s comment, we have added the following to the main text:

“In Mechanical Engineering, there are metamorphic mechanisms and kinematic based metamaterial systems that can achieve a large number of different configurations (>10) through kinematic shape morphing (see Fig. 1D) [9, 10, 11, 12]. However, existing research has not shown whether these metamorphic systems have a good load-carrying capability to serve as civil structures. Furthermore, many of the configurations offered from such morphing systems do not provide useful shapes such as columns, beams, bridges, and walls that are needed for civil structures.” (Page 3)

We have updated Fig 1 and main text to further address the comment.

In addition, **Table S2** is added to the supplementary material to introduce metamorphic and reconfigurable mechanisms. The new supplementary text in section S1 also presents an additional literature review to systematically summarize all relevant technologies in the field of civil engineering, aerospace engineering, and mechanical engineering.

As a summary, the following parts in the manuscript and SI have been revised:

Manuscript: **Fig. 1**. Text on **Page 3**

SI: **Supplementary Text Section S1** and **Table S2**.

Reviewer #3 (Remarks to the Author):

This paper presents the Modular and Uniformly Thick Origami (MUTO) system for large-scale, load-carrying structures. The paper makes important contributions to origami geometry for large-scale applications. However, the authors should take into consideration the following comments, particularly related to the experimental investigation.

Reply: Thank you for giving a high evaluation of this work. We believe your suggestions have significantly improved the quality and clarity of this work. We have improved the paper based on your review comments. More specifically, additional discussion and information is provided to strengthen the section related to the experimental investigation.

1. The statement that “Existing civil engineering structures cannot change their shape and adapt for new functions...” is incorrect. It ignores a long history of civil engineering structures that are modular, deployable, and rapidly erectable. The paper would benefit from a more rigorous literature review that includes both buildings/shelters and bridges.”

Reply: Thank you for this important comment. It was not our intention to ignore or dismiss the historical use of modular and deployable structures in civil engineering. Instead, we believe the proposed MUTO systems demonstrate an entirely different philosophy for adaptability, deployment and modularity when compared with existing systems. We have revised the paper and as suggested we have presented a more thorough literature review to better compare and contrast our work with the current state-of-the-art.

Regarding the specific statement, we agree with the reviewer that our word choice of “cannot” was not precise – we have now changed the wording to:

“Existing Civil Engineering structures have limited capability to adapt their configurations for new functions, non-stationary environments, or future reuse.” (**Abstract, Page 1**)

We believe the updated sentence is a fairer statement regarding the state-of-the-practice of existing civil systems. We have also improved the language throughout the paper to avoid overstatements like the one mentioned in this comment.

Next, in the supplementary section S1, we have added a literature review and more thorough discussion on existing modular, deployable, and rapidly erectable systems (**Tables S1 to S4**). This added material better summarizes the state-of-practice technologies and provides a direct comparison with the proposed MUTO system. The MUTO’s capability to drastically change its shape (from a bridge to a bus stop) is beyond the capability of common erectable and deployable civil systems. The literature review is summarized in the new **Fig. 1** of the manuscript.

As a summary, the following parts in the manuscript and SI have been revised:

Manuscript: Text on **Page 1**; **Fig. 1**

SI: Added **Supplementary Text Section S1**. Added **Table S1 to S4**

2. The statement “Origami with uniform thickness can transfer compression and tension forces without producing moments (Fig. 2B) which in turn enables better load-carrying performance.” is quite strange as the author later use origami in a bending environment (with flexural connections). Please clarify.

Reply: Thank you for pointing out this potential point of confusion. First, we believe there was potentially some misunderstanding. We have rewritten this sentence to clarify more explicitly.

“Origami with uniform thickness can transfer compression and tension forces without producing moments at the connection point after the crease is closed using locking devices (Fig. 2C). In contrast, axial forces within origami with non-uniform thickness will result in unwanted moments that could limit the load-carrying performance. Moreover, having uniform thickness also enables better bending capacity, because straight connection plates have better strength and stiffness when compared to zig-zag plates (Fig. 2C).”

Note that Fig. 2B is now 2C. (Page 6)

Moreover, we agree with the reviewer that our connections can also be used in a global and/or local bending scenario after they are locked. In this situation, having uniform stiffness still produces better capacity, because the connection plates can be made straight. If the closed connection (shown in Fig 2C) is taking a bending moment locally, the upper connector plate needs to transfer a local tension or compression force. Having a zig-zag connector plate would be needed for a non-uniformly thick origami. This plate will bend under tension or compress forces due to the applied bending.

Furthermore, we have added a new **Supplementary Video SV2** to show how locking devices can alter the system from a mechanism to a structure capable of supporting bending loads or vice versa. We believe the revised text and new supplementary video can avoid confusion.

As a summary, the following parts in the manuscript and SI have been revised:

Manuscript: Text on **Page 6**.

SI: **Supplementary Video SV2**

3. The connections (Fig. 3) are the most important components for the performance of large-scale structures. Providing information on the connection design, how it meets current codes (both bridge and building), and how it performs during the experiments would be critical for the paper.

Reply: Thank you for this comment. Yes, connections are significant for these and for large-scale structures in general. We have rewritten the supplementary material to give more details regarding the design and performance of these connectors. Here is a list of edits we made to the main text and the SI:

- (1) We have included **Section S6** and two additional **Figures S12** and **S13** to demonstrate the design details of different connectors for the MUTO bridge. The two figures are shown below:

- (2) We have shown a detailed calculation to find the nominal capacity of these connector plates. In this calculation, we showed that the connection plate will not yield under the ultimate load of this structure (our experiment shows 2500N at the mid span). We have checked the following capacity:
- Bearing capacity of screws (capacity/load = 1.52) (**Fig S15D**)
 - Tensile strength of Al connector plate (capacity/load = 9.4) (**Fig S15D**)
 - Tensile strength of MDF (capacity/load = 3.06) (**Fig S15D**)
 - Shear capacity of M3 bolt (capacity/load = 1.58) (**Fig S15D**)
 - Bending strength of angled Al connector plate (capacity/load = 1.03) (**Fig S15E**)
 - Shear strength of angled Al connector plate (capacity/load = 18.6) (**Fig S15E**)

This calculation can be found in **Section S7** of SI. The following **Fig. S15** is added to the SI to illustrate the calculation process. These calculations follow general procedures for computing the nominal the capacity of heavy timber structure connectors (Such as the GB/T 50708-2012 Chinese Technical Code of Glued Laminated Timber Structures)

- (3) We have included a more detailed documentation of the sliding behavior in gusset plate connectors. The following Fig. S16 is added to the SI Section S7.

As a summary, the following parts in the manuscript and SI have been revised:
 SI: Text in Section S6 and S7; Fig. S12, S13, S15, and S16.

4. In the videos, the connections require a high degree of manual labor. This is a weakness in the MUTO system as compared to the state-of-the-practice in modular and rapidly erectable structures. Please provide further discussion.

Reply: Thank you for this comment. Yes, using gusset plates does require a relatively high degree of manual labor. However, we believe the labor presented in our videos is not drastically different compared to other types of construction that use gusset plates for connection (e.g. heavy timber and steel structures). Furthermore, a partial benefit of the MUTO design even with gusset plates is that many of the connection points are pre-aligned and conveniently held in place by the opposing hinges. Regardless, the issue related to a high degree of manual labor can be addressed by using better connectors such as the latch locks and sliding locks presented in this work.

In the updated manuscript, we provided additional experiments comparing the time needed to assemble a MUTO bridge using only gusset plates versus using a combination of latch locks and gusset plates. Our results indicate that the assembly process using latch locks can be four times faster than using gusset plates. These results are demonstrated in **Supplementary Video SV3** and **SV6**. With these results, we believe future research can further improve the MUTO system by designing better and faster to assemble connectors. For example, we believe that the Self-latching locks concept presented in Fig 4D can provide an even faster assembly. To address the comment, the following text was added to the manuscript:

“The design of the individual connectors can significantly affect the assembly speed of entire the MUTO system. In Supplementary Video SV3, the MUTO bridge uses 16 sliding locks, 78 latch locks, and 8 gusset

plates for assembly. With two people, this bridge is assembled in 15 minutes, or a total of 30 minutes worker time. In Supplementary Video SV6, we provide a comparison where the same MUTO bridge is assembled using 16 sliding locks and 100 gusset plates. In this scenario, the assembly process requires a total of 120 minutes worker time. This result indicates that using rapid locking hinges (such as latch locks, sliding locks, or self-latching locks) can significantly improve the assembly speed and potentially lower the labour cost. A discussion on the construction of these two MUTO bridges can be found in Supplementary Text Section S6.” (Page 12)

Furthermore, we do not think that the MUTO bridge assembly approach is necessarily weaker than state-of-the-practice in modular and rapidly erectable structures. For instance, the deployable housing shelters shown in (Thrall and Quaglia, 2014, *Engineering Structures*, 59, 686-692) require 5-10 soldiers to assemble and the reported deployment time can be from 30 minutes to a day depending on the design. The military scissor bridge such as one shown in (Thomas and Sia 2013 “A Rapidly Deployable Bridge System,” *Structural Congress*) can be deployed with less labor through using hydraulic actuators. Using actuators is ok for military applications but can be too expensive for commercial civil applications. Assembling inflatable buildings may also require less labor than using MUTO, but these systems provide limited load-carrying stiffness and have just two configurations. (These existing systems are discussed in the new **Supplementary Table S1 to S4**).

As a final note, while we do acknowledge that the MUTO system can be improved in future research, we believe it still provides unique advantages when compared with state-of-the-practice techniques summarized in the **Supplementary Text Section S1**.

As a summary, the following parts in the manuscript and SI have been revised:

Manuscript: Text on **Page 12**

SI: Supplementary Text **Section S1**, Supplementary **Table S1 to S4**, Supplementary **Video SV6**

5. The authors have a section titled “High Load-Carrying Capability.” How is “high” quantified and how does it compare to a similar structure that does not use origami?

Reply: Thank you for raising this question. We agree with the reviewer that this statement may cause confusion. By using “High”, we are qualitatively comparing our thick origami systems with thin origami systems. To clarify this, we added the following sentence in the main text:

“This section shows that thick origami can achieve much higher strength and stiffness when compared to their thin-origami counterparts [25].” (Page 13)

After adding in thickness, we can make the proposed MUTO system as strong as and as stiff as a structure that does not use origami by making the cross-sections smaller or larger. However, we also acknowledge that when comparing MUTO to other structures, load-carrying performance alone is not a fair point of comparison - it may be better or worse depending on how/what we compare against.

We believe a more holistic comparison that includes load-carrying, amount of material used, and other benefits of the MUTO need to be considered. For example, while we know that MUTOs will require more material to reach the same stiffness and strength of a static counterpart, they offer geometric adaptability, faster construction, and a capability for reuse, that may ultimately result in lower life-cycle carbon emissions. Comprehensive studies like the life-cycle analyses are beyond the scope of this work.

However, we do believe this is fruitful future research and have added the following sentence to the discussion:

“Future research can also study the embodied carbon and life-cycle cost of adaptable MUTO systems when compared to traditional non-reusable systems and/or other adaptable structural solutions. These studies will provide guidelines to engineers regarding how to pick a more appropriate structural system for a specific design scenario.” (Page 15)

As a summary, the following parts in the manuscript and SI have been revised:
Manuscript: Text on Page 13 and Page 15

6. The experimental aspect of the paper, and comparison with the modeling, requires major revision. Areas that should be addressed:

Reply: Thank you for the following suggestions, we have addressed the reviewer’s comments, which have substantially strengthened both the experimental aspects and modeling comparisons in our work.

-Behavior of the MUTO system is not compared to a “control” specimen (e.g., a similar system that cannot deploy). This is needed to put the later discussion into context for the readers.

Reply: We appreciate this suggestion from the reviewer. We agree that comparing to a ‘non-deployable control’ system would be useful when performing a holistic study considering including life-cycle cost and carbon emissions (see response to comment # 5 above. However, we believe that comparing strength and stiffness between these systems will not add value to our current work for the following reasons:

- 1) We could potentially fabricate a structure of the same size and topology but made of continuous wood and without any connections. We know that such a control system would be stiffer and likely stronger than our MUTO prototype.
- 2) As discussed in our response to comment #5 above, we believe a MUTO system can be designed to match the strength and stiffness of a traditional non deployable civil structure by enlarging the cross section and using more conservative connectors. The MUTO design will simply require more material usage.
- 3) A fair comparison of such systems would require evaluation of the other benefits of MUTOs such as the overall life-cycle costs, carbon emission, faster assembly time, less onsite perturbation. However, fully addressing these items is beyond the scope of this work.

Thus, to address the comment by the reviewer, we have added the following sentence to the manuscript when discussing the load-carrying capability of MUTO systems:

“By switching the material to structural steel (or other materials) and enlarging the cross-sections, our MUTO can be made to be as stiff as and as strong as non-deployable civil structures.” (Page 14)

“Future research can also study the embodied carbon and life-cycle cost of adaptable MUTO systems when compared to traditional non-reusable systems and/or other adaptable structural solutions. These studies will provide guidelines to engineers regarding how to pick a more appropriate structural system for a specific design scenario.” (Page 15)

Furthermore, although a comparative study on the strength and stiffness may not add great value to this work, we do find that a comparative study on the connector types can be important. We have included a comparative study regarding the effects of different connectors on the assembly performance of MUTO systems. (see **Supplementary Video SV3** and **SV6**). In this comparative study, we compare the assembly speed between one gusset plate only prototype and one latching hinge enhanced prototype. We showed that using fast hinges like latch lock can significantly increase the assembly speed of MUTO systems. This comparison shows the importance of connector design for deployable structures – it significantly affects both the assembly performance and the load carrying performance of MUTO systems.

As a summary, the following parts in the manuscript and SI have been revised:

Manuscript: Text on **Page 14** and **Page 15**

SI: **Supplementary Video SV6**

-The experimental test setup needs to be described in far greater depth (even in the supplemental material). For example, what are the boundary conditions?

Reply: Thank you for raising this question, we have rewritten the supplementary material to give a more detailed introduction of the experimental set up. The bridge is simply supported using a support built with 2 by 4 lumber. There is no direct connection between the supporting structure and the bridge, so the bridge can freely deform at the ends. The support deformation is also monitored using the Optotrack system and the minimal deformations of the supports are appropriately subtracted when calculating the mid-span displacement. The following text is added to the manuscript:

“Figure 7A shows the experimental setup for the 2-metre-long MUTO bridge where the bridge is simply supported at the two ends. We applied a displacement-controlled cyclic loading scheme to test the MUTO bridge where the actuator applies incremental deformations at the mid span.” (**Page 13**)

In addition, we add the following **Fig. S14** to explain the details:

For the MUTO column, it is first placed between two distribution wood plates and then put under the hydraulic loading machine. The column is centered under the hydraulic actuator load cells before the experiment. There is no oil or lubricant used at the end of the column to mitigate friction. Details are shown in the updated **Fig S18** of SI.

In addition, the following text is included in the manuscript to introduce the boundary condition set of the MUTO column experiment:

“The column is first placed between two distribution plates and then put under the actuator, where the column is directly loaded to failure.” (**Page 14**)

The bottom of the column cannot rotate. However, the top end can experience small rotation when the actuator presses downward. This is because the top connector between the actuator and the end plate is a rotational hinge (see updated Fig. S18).

As a summary, the following parts in the manuscript and SI have been revised:

Manuscript: Text on **Page 13** and **Page 14**

SI: **Fig. S14 to S18**, **Supplementary Text Section S7 and S8**

-Stresses from simulation are compared to strains, and are stated as having “good agreement”, but no quantitative comparison is made. What assumptions are being made in comparing stresses to strains? How close are they?

Reply: Thank you for the question. We are comparing strain to strain. The strain from the simulation can be calculated by dividing the obtained stress with the Young’s modulus of the system. We do notice that the original Fig. 6 (*now Fig. 7*) did not have a clear legend, so we believe the reviewer may have missed the bold line in the **Fig 7D** which quantitatively shows the comparison. We have updated the figure to avoid confusion. Please see the updated figure below:

As a summary, the following parts in the manuscript and SI have been revised:
Manuscript: **Fig 7**.

-The peak loads are not discussed within any context (see comment above as well).

Reply: Thank you providing this comment. Yes, peak load is one important item to discuss. We have included additional discussion on the peak load for both the bridge and the column.

“Using the simulated relationship between the maximum stress and the applied load (compressive $\sigma_{max} = 4.8 \text{ MPa}$ at 1.5 kN load) and the compressive strength of MDF (10MPa), we can analytically estimate that the ultimate failure load of the bridge is 3.13 kN (based on material failure). The bridge fails at 2.5 kN because buckling of the compressive truss introduces higher stress in the member than the theoretical calculation. Still, the bridge is able to achieve 80% of its ultimate capacity before the instability failure.”
(Page 13)

“We can show that the theoretical ultimate force capacity of this column is 20.5 kN using the compressive strength of MDF, which is 10 MPa (see Supplementary Text Section S8). We can further confirm that material failure is the leading cause of failure with the recorded video of the experiment, where we found that the onset of failure is marked by delamination of MDF trusses (see Supplementary Text Section S8 and Supplementary Video SV8).” (Page 14)

The added discussion shows that buckling instability contributed to the failure of the bridge but the column failure is predominantly a material failure.

As a summary, the following parts in the manuscript and SI have been revised:
Manuscript: Text on **Page 13** and **Page 14**
SI: Added discussion on **Section S7** and **S8**

-For the bridge, a mid-span displacement of 1/125 of the span is discussed. In typical bridge practice, the limit is 1/800 of the span. Please consider the displacement in context with current practice. Similar comments for the column.

Reply: Thank you for this question. We want to point out that the mid-span displacement limit pointed out by the reviewer (1/800) is not directly comparable to our reported value. The displacement ratio of 1/125 is **at actual failure** but the 1/800 is the **serviceability** limit of bridges (at a lower loading level). For example, please see the following reference, where a real highway bridge is tested to failure. (Scanlon & Mikhailovsky, 1986, Full-scale load test of three-span concrete highway bridge, *Can. J. Civ. Eng.* 14, 19-23). In their experiment, the 18.6m mid span is loaded to a maximum displacement of 157mm, which is a mid-span displacement of 1/116. At this displacement level, the bridge was seen to be in the failure state. With this number, we believe the demonstrated MUTO bridge has a behavior that is close to realistic civil structures.

[REDACTED]

To address this in the manuscript, we added the following sentence to the main text:

“For example, in a real-scale testing of a high-way bridge, a mid-span displacement ratio of 1/116 was recorded at failure [54].” (Page 13)

As a summary, the following parts in the manuscript and SI have been revised:
Manuscript: Text on Page 13

-Important quantities like stiffness should be discussed.

Reply: Thank you for this comment. Yes, stiffness is an important characteristic of structures. We have added additional discussion of stiffness for both the MUTO bridge and MUTO column design. The following text is added to the main text.

“For example, in a real-scale testing of a high-way bridge, a mid-span displacement ratio of 1/116 was recorded at failure [54]. The initial stiffness of this bridge is 280 kN/m, which is much larger than thin-origami systems built at this scale [25].” (Page 14)

“This column has an initial stiffness of 1875 kN/m. At the peak force, the column experiences 10 mm of displacement at the top, which is reasonably stiff for a column built with soft MDF material and a small cross-section area of just 20.5 cm².” (Page 14)

In addition, we have added some discussion on how we can potentially increase the stiffness and strength of the MUTO by enlarging the cross section and switching to other structural materials.

“Although testing MUTO systems built with different materials may eventually reveal other unforeseen failure modes, we believe that the MUTO concept can achieve comparable stiffness and strength when compared to non-deployable civil structures. By switching the material to structural steel (or other materials) and enlarging the cross-sections, MUTOs can be made to achieve a comparable structural performance to non-deployable civil structures.” (Page 14)

As a summary, the following parts in the manuscript and SI have been revised:
Manuscript: Edited text on Page 14

-One of the most interesting aspects of the experimental program is the sliding of the connectors. Please add further discussion.

Reply: Thank you for this comment. Yes, we agree that the sliding behavior is important for understanding the behaviors of the MUTO. We have added the following **Fig S17** in the supplementary text to explain why there is sliding in the connector. We used standard bolts to transfer loads at the connection. In this connection, the bolt can slide inside the bolt hole, which results in hysteretic behavior. We believe this sliding can be mitigated by using high strength bolts and a friction connection. A further discussion on the sliding behavior is now included in the supplementary information.

“This sliding behavior occurs in the connectors because we use standard bolts. When using standard bolts, there is limited friction between the gusset plate and the connected MDF member. When applying cyclic loading, there will be sliding between the gusset plate and the MDF member, which produces hysteresis. Figure S17 shows how slacks between bolts and connectors can produce this hysteresis. When tension is applied, gaps will form in the connectors until the bolts come into contact with connection plates and MDF; when compression is applied, gaps will be closed and there is direct contact between MDF. In Civil Engineering practice, we can replace these standard bolts with high-strength bolts to prevent sliding behaviors. High-strength bolt can produce a high friction force, which stops the relative motion between the gusset plate and the wood member. Another way to stop the relative motion is to use wood screw rather than the M3 machine screws. However, using wood screw may cause the structure to be no longer reusable.”

As a summary, the following parts in the manuscript and SI have been revised:
SI: **Figure S17**. Text to **Page 32**.

-“Material failures” are identified. However, it is not clear if the material strength is the issue as the failures seem to be occurring at points where there is added thickness between layers. Further investigation of this is necessary.

Reply: Thank you for raising this critical comment. Yes, we agree with the reviewer that it was necessary to take a closer investigation at the failure mode of this MUTO column. *In conclusion, material failure is indeed what occurs in the system.* The MDF material has a compressive strength of 10 MPa, the column has an ultimate load of 21 kN and an effective cross-section of 20.5cm². The total capacity of the column is $10MPa \times 20.5cm^2 = 20.5 kN$ which is remarkably close to the measured ultimate force of the column. Thus, we believe material failure is the cause. Furthermore, in the recorded video of the experiments, we can see that the onset of failure is marked by MDF delamination, which is what we would expect for a

compressive material failure (see **Fig S18** below). We have updated the **Supplementary Video SV8** to highlight the failure occurrence. The additional calculation can be found in **SI Page 34**.

There is a reason why material failure occurs at the location. The weakest spot is located close to where there is extra thickness (see **Fig S18** below). The red box indicates the weakest spot, the blue box indicates location with extra thickness, while the green box indicates the structural hinge. The structural hinge will restrain the MDF bar, so the weakest place is at the location indicated with red boxes. This figure and additional text are added to the supplementary information to further clarify the behavior.

As a summary, the following parts in the manuscript and SI have been revised:
 SI: **Fig. S18**. Text to **SI Page 34**. **Supplementary Video SV8**

7. The authors are making comparisons to other civil engineering materials on the basis that there was an MDF material failure (see comment above). However, these are all very different materials with very different behaviors. This comparison is not appropriate as it is currently performed. Instead, the authors should model each of these types of structures and compare performance, as well as weight.

Reply: Thank you for raising this critical suggestion. We agree with the reviewer that the different materials give different behaviors. Thus, if material failure is not the major cause, the comparison may not be appropriate. To address this, we have introduced a factor F to evaluate if stability may cause failure after we change the material. Assuming that the structure geometry is not changed, we know that the buckling load $P_{cr} = \pi^2 \frac{EI}{L^2}$ scales with the Young's modulus of material while the material failure load $P_u = \sigma A$ scales with the material strength. Then, we can use the following F factor to evaluate if stability failure may occur or not.

$$\frac{P_{cr}^A}{P_u^A} = F \frac{P_{cr}^{MDF}}{P_u^{MDF}} = \frac{P_{cr}^{MDF} \left(\frac{E_A}{E_{MDF}} \right)}{P_u^{MDF} \left(\frac{\sigma_A}{\sigma_{MDF}} \right)} = \frac{P_{cr}^{MDF}}{P_u^{MDF}} \frac{E_A}{E_{MDF}} \frac{\sigma_{MDF}}{\sigma_A}$$

$$F = \frac{E_A}{E_{MDF}} \frac{\sigma_{MDF}}{\sigma_A}$$

Because we know that the MDF column is controlled by material failure, $\frac{P_{cr}^{MDF}}{P_u^{MDF}} > 1$. Then, if the above factor F is also greater than one, we know that material failure will still likely be the major cause of failure after we change the material (see detailed derivation in the new SI section S10 **Page 38**). With this additional analysis, we know that after changing material to structural steel and UHP, material failure would still likely be the major cause. However, for aluminum and CLT, because they have smaller Young's modulus, it is possible that a stability failure can occur and change the failure mode. Therefore, these data points are now displayed in grey, indicating that they may be affected by other failure mechanisms. The following text is also added to the main text to clarify our assumptions and limitations of the extrapolation:

“When performing this extrapolation, we assume that the column fails because of material failure only with no instability issues. We also assume that the structure maintains the same geometry and only the material properties (Young's modulus E , compressive strength σ , and density ρ) are changed. Our analysis in Supplementary Text Section S10 shows that after switching the material from MDF to steel and UHPC the MUTO column should still fail due to material failure. However, when switching the material to CLT and Aluminium, the column may fail due to an instability. In this case the capacity can be smaller than what is shown in Fig. 7G (indicated with grey colour). Although testing MUTO systems built with different materials may eventually reveal other unforeseen failure modes, we believe that the MUTO concept can achieve comparable stiffness and strength when compared to non-deployable civil structures. By switching the material to structural steel (or other materials) and enlarging the cross-sections, MUTOs can be made to achieve a comparable structural performance to non-deployable civil structures.” (**Page 14**)

As a summary, the following parts in the manuscript and SI have been revised:

Manuscript: Text on **Page 14**

SI: Text to SI **Page 38**.

Reviewer #4 (Remarks to the Author):

This paper presents a highly novel type of deployable structure, based on the use of modular, thick-panel, and superimposed folding patterns. The authors have identified a clear problem in the scale-up of origami-inspired structures, and their proposed 'MUTO' system is original and demonstrated to provide a viable solution. The findings are of high significance for the structural origami research community.

The paper is in general well written, with the figures and supplementary videos in particular helping the reader to follow the theoretical and practical advancements of the MUTO system. The following items are raised as questions for minor clarification.

Reply: Thank you for giving a high evaluation of this work. We believe your suggestions have significantly improved the quality and clarity of this work. We have improved the paper based on your review.

Comments/Questions:

INTRODUCTION SECTION

- Fig 1 - Is the packaging ratios for the MUTO system calculated anywhere in text or in supplementary materials? To what extent is this influenced by the panel thickness parameter t .

Reply: Thank you for raising this question. The packaging ratio can be calculated based on the packed volume and the deployed volume. This calculation is now shown in the updated supplementary text section S1. The following **Fig. S3** is added to explain the relationship between the packing ratio and the panel thickness.

As a summary, the following parts in the manuscript and SI have been revised:
SI: **Fig. S3**. Text on **Page 7**.

- The 'number of configurations vs packaging ratio' is a novel performance metric, however what is meant for the 'modular civil structures' category shown in Fig 1? There are some kit-of-part type systems which would fall into this class and have #configurations $\gg 1$, albeit they would not necessarily be deployable structures.

Reply: Thank you for raising this point. When discussing modular civil systems, we primarily referred to cargo container type buildings and prefabricated modular construction. We have updated Fig. 1 to make this clear and avoid miscommunication.

Furthermore, the reviewer is correct and there are some kit-of-part type systems that can achieve multiple configurations. As the reviewer suggests, these systems use a different philosophy of design and as such are not necessarily deployable. We have further mentioned these systems in the following sentence:

“There are also prefabricated “kit-of-parts” structural systems that can form large structures by manually assembling small individual members on site [5]. However, each structural member is not deployable which constrains its size to a pre-specified transportation limit.” (Page 3)

We believe this sentence can highlight the existence of these kit-of-part type systems, while also pointing out the deployment capability of the MUTO.

In addition, we included a literature review in the supplementary **text section S1**. In this literature review, we have systematically summarized related deployable structures from Civil Engineering, Aerospace Engineering, and Mechanical Engineering using **Table S1 to S4**.

As a summary, the following parts in the manuscript and SI have been revised:

Manuscript: **Fig. 1**. Text on **Page 3**.

SI: **Supplementary Table S1 to S4**. Added Supplementary Text **Section S1**.

- pg 2, ln 26-30 low/high capacity to support structural loads, 'softer and lighter' - these terms seem a little bit vague. Are you referring to relaxed or n/a serviceability requirements for space structures?

Reply: Thank you for bringing this up. We agree that the terms are vague. What we mean is that space systems are in zero-gravity environment, so they do not need to support loads imposed by gravity including the support of self-weight. As a comparison, self-weight already consumes much of the structural capacity for typical civil structures. We have rewritten the sentence to avoid misunderstanding.

“However, these systems are used in zero-gravity conditions, so their design is different from civil structures where self-weight can make significant contributions to the loading of a structure.” (Page 3)

As a summary, the following parts in the manuscript and SI have been revised:

Manuscript: Text on **Page 3**.

LOCAL AND GLOBAL MODULARITY

- how many distinct modular panel types are used in the demonstrated prototypes? The paper mentions the 'solid' and 'triangular' panel types, but it is not clear if panel handed-ness or crease polarity makes these all the same or with a few different versions of each.

Reply: Thank you for bringing this up. Yes, we agree with the reviewer that handed-ness can be significant for the construction of MUTO. However, we have avoided the handedness issue in our fabrication by putting symmetric connection holes on both side of the panels. In this case, holes on one side of the panel are not used. With this approach, every truss panel is the same and every solid panels is the same. We have added this sentence to the main text:

“When fabricated with symmetric connection holes on both sides, the triangular panels are not subject to handedness and can be installed in any matching orientation.” (Page 9)

Furthermore, we have included a laser cutter file to further illustrate this design philosophy.

As a summary, the following parts in the manuscript and SI have been revised:

Manuscript: Text on Page 9.

SI: Added Supplementary Cutter File

NAVIGATING BIFURCATION

- the modelling conducted for the bifurcation transition path assumes ALL creases of a particular type/subset are locked to support the lowest-energy transition to the desired state. Is locking all creases within a particular subset necessary, or will locking some/most of them sufficient?

Reply: Thank you for asking this interesting question. Yes, we assume that all creases of a specific subset are locked to support the transition, which is probably what would be done in practice. However, it is true that similar transitions can be achieved while not locking all of the indicated creases. We believe this is an exciting future research direction to better understand the behavior of MUTO systems. We have added the following sentence to the main text:

“With the simulation capability, future research can explore the effects of locking a subset of creases instead of all designated creases shown in Fig. 6F. Such an investigation can help us understand whether individual creases are more critical for the transition between different kinematic paths.” (Page 12)

As a summary, the following parts in the manuscript and SI have been revised:

Manuscript: Text on Page 12

REVIEWER COMMENTS

Reviewer #1 (Remarks to the Author):

The authors have addressed my comments well and substantially improved the manuscript. I think this is ready to be published.

Reviewer #2 (Remarks to the Author):

Comments on "Modular and Uniformly Thick Origami for Large-Scale, Load-Carrying, and Adaptable Structures"

The authors have addressed some of the reviewer's comments which improved the quality of the manuscript. Due to the lack of significant innovation in the design of the origami structure, the manuscript has yet to reach the high quality of publication in Nature Communications.

First, one of the main achievements of this manuscript is to propose developable, flat foldable, and uniformly thick origami vertices of degree-6 and degree-10. Vertex of degree-6 is a diamond-shaped vertex and has been extensively studied before. And there is no further investigation on vertex of degree-10.

Second, the author's minor modification in origami structural design is using superimposition creases, resulting in a single-degree-of-freedom origami system becoming multi-degree-of-freedom. This undesirable reconfiguration mechanism in this work requires a large number of locking devices when switching between different configurations, and these locking mechanisms are time- and labour-intensive, as shown in their videos. The authors explained in their reply that the purpose of superimposition crease was to allow obtaining a structure with arbitrary aspect ratios while maintaining compact packaging, however, there is no prominent role in the text for the structure of these arbitrary aspect ratios.

Next, the authors misuse the term "kinematic bifurcation", Please see [R1]- [R3].

[R1] Tarnai, T. (2001). Kinematic bifurcation. In *Deployable Structures* (pp. 143-169). Vienna: Springer Vienna.

[R2] Dai J S, Rees Jones J. Mobility in metamorphic mechanisms of foldable/erectable kinds. *Journal of Mechanical Design*, 1999, 121(3): 375-382.

[R3] Zhang, L., & Dai, J. S. (2009, June). An overview of the development on reconfiguration of metamorphic mechanisms. In *2009 ASME/IFTOMM International Conference on Reconfigurable Mechanisms and Robots* (pp. 8-12). IEEE.

Finally, there are quite a few unproven conclusions. Examples are not limited to those listed below: (1) "we show that uniform thickness of MUTO systems enables them to carry large loads." There is no direct evidence that the load-carrying capacity comes from uniformly thick-panel, it seems more likely that it comes from the locking devices.

(2) "we have observed two trends: first, origami structures that can fold cannot carry large loads [25, 26], and second, origami structures that can carry large loads cannot fold [27, 28, 29].

Moreover, common origami patterns for civil applications tend to have only one kinematic path [30, 25], so they are not well suited for creating adaptable systems that offer multiple configurations." The authors have missed some of the origami research that are foldable and capable of bearing utilizing self-locking or bifurcation ([R4]-[R9]).

[R4] Ye, H., Liu, Q., Cheng, J., Li, H., Jian, B., Wang, R., ... & Ge, Q. (2023). Multimaterial 3D printed self-locking thick-panel origami metamaterials. *Nature Communications*, 14(1), 1607.

[R5] Jamalimehr A, Mirzajanzadeh M, Akbarzadeh A, et al. Rigidly flat-foldable class of lockable origami-inspired metamaterials with topological stiff states[J]. *Nature communications*, 2022, 13(1): 1816.

[R6] Zhao, Z., Kuang, X., Wu, J., Zhang, Q., Paulino, G. H., Qi, H. J., & Fang, D. (2018). 3D printing of complex origami assemblages for reconfigurable structures. *Soft Matter*, 14(39), 8051-8059.

[R7] Zhai, Z., Wang, Y., & Jiang, H. (2018). Origami-inspired, on-demand deployable and collapsible mechanical metamaterials with tunable stiffness. *Proceedings of the National Academy of Sciences*, 115(9), 2032-2037.

[R8] Tomita, S., Shimanuki, K., Nishigaki, H., Oyama, S., Sasagawa, T., Murai, D., & Umemoto, K.

(2023). Origami-inspired metamaterials with switchable energy absorption based on bifurcated motions of a Tachi-Miura polyhedron. *Materials & Design*, 225, 111497.

[R9] Yasuda H, Gopalarethinam B, Kunimine T, et al. Origami-based cellular structures with in situ transition between collapsible and load-bearing configurations[J]. *Advanced Engineering Materials*, 2019, 21(12): 1900562.

(3) The author's reply "Our work ...explores modularity of MUTOs to achieve high adaptability, harnesses kinematic bifurcations for system reconfiguration, and uses experimental tests to demonstrate the high strength and stiffness of these structures. These characteristics were not previously demonstrated in the field of structural origami." The previous works in [R10-R14] with kirigami, or modular origami techniques are essentially folding something using the reconfigurability of multi-DOFs systems or kinematic bifurcation of a single DOF system.

[R10] Li Y, Zhang Q, Hong Y, et al. 3D transformable modular Kirigami based programmable metamaterials. *Advanced Functional Materials*, 2021, 31(43): 2105641.

[R11] Li Y, Yin J. Metamorphosis of three-dimensional kirigami-inspired reconfigurable and reprogrammable architected matter. *Materials Today Physics*, 2021, 21: 100511.

[R12] Yamaguchi K, Yasuda H, Tsujikawa K, et al. Graph-theoretic estimation of reconfigurability in origami-based metamaterials. *Materials & Design*, 2022, 213: 110343.

[R13] Liu W, Jiang H, Chen Y. 3D programmable metamaterials based on reconfigurable mechanism modules[J]. *Advanced Functional Materials*, 2022, 32(9): 2109865.

[R14] Wang C, Li J, Zhang D. Motion singularity analysis of the thick-panel kirigami[J]. *Mechanism and Machine Theory*, 2023, 180: 105162.

Reviewer #4 (Remarks to the Author):

The authors have satisfactorily addressed all reviewer comments.

Revisions and response to referees' comments on Manuscript NCOMMS-23-39886A entitled: "Modular and Uniformly Thick Origami for Large-Scale, Adaptable, and Load-Carrying Structures"

by Y. Zhu and E.T. Filipov.

We are grateful to the reviewers for their feedback and helpful critique which have now brought about greater clarity to the paper. The reviewers' comments have been addressed in the manuscript, and the revisions are summarized below. Sections of the text that have been modified have been colored blue in both the manuscript and the supplementary text.

Reviewer #2 (Remarks to the Author):

The authors have addressed some of the reviewer's comments which improved the quality of the manuscript. Due to the lack of significant innovation in the design of the origami structure, the manuscript has yet to reach the high quality of publication in Nature Communications. First, one of the main achievements of this manuscript is to propose developable, flat foldable, and uniformly thick origami vertices of degree-6 and degree-10. Vertex of degree-6 is a diamond-shaped vertex and has been extensively studied before. And there is no further investigation on vertex of degree-10. Second, the author's minor modification in origami structural design is using superimposition creases, resulting in a single-degree-of-freedom origami system becoming multi-degree-of-freedom. This undesirable reconfiguration mechanism in this work requires a large number of locking devices when switching between different configurations, and these locking mechanisms are time- and labour-intensive, as shown in their videos. The authors explained in their reply that the purpose of superimposition crease was to allow obtaining a structure with arbitrary aspect ratios while maintaining compact packaging, however, there is no prominent role in the text for the structure of these arbitrary aspect ratios.

Reply: First, we thank the reviewer for taking the time to review our manuscript. We are glad that we have addressed their comments, and have improved the quality of the manuscript. *However, we continue to strongly disagree with the comment that our work lacks significant innovation.* The Modular and Uniformly Thick Origami (MUTO) can deploy into meter-scale structures, adapt into different structural forms, and carry remarkably large loads (hundreds of kilograms to tons). Our work derives necessary conditions for developable and flat foldable thick origami, explores modularity of MUTOs to achieve high adaptability, harnesses multi-path folding motions for system reconfiguration, and uses experimental tests to demonstrate the high strength and stiffness of these structures. These characteristics were not previously demonstrated in the field of structural origami. We believe the significance is also reflected by the high evaluations from the other three reviewers.

Regarding the reviewer's more specific comments: The achievement of this work is not just finding one degree-6 and one degree-10 vertex to achieve adaptive civil structures. We have derived necessary conditions of developability and flat foldability in a generic degree-N vertex, as shown in the updated Figure 2 of this paper. These theoretical development will enable future exploration of generic degree-N thick origami vertex. Next, the work introduces a bar-and-hinge simulation method, and multiple locking hinge designs to navigate the multi-path deployment process of the proposed MUTO pattern. With these locking devices, we can easily convert MDOF mechanism back to SDOF systems and control their deployment paths. We disagree that this is in any way a detriment to our system. Additionally, our literature review has demonstrated that the proposed method can offer comparable or even faster deployment when compared to deployable civil structures at meter scale. Moreover, the adaptability of the proposed system is not presented in both state-of-the-practice designs and state-of-the-art research of deployable civil-scale structures. Finally, the crease superimposition does indeed play a prominent role in the demonstrated structures. Both

the two meter long bridge and the four meter long bridge rely on the specific superimposition to obtain large aspect ratio (length over width). The standard Yoshimura pattern cannot achieve these configurations.

Next, the authors misuse the term “kinematic bifurcation”, Please see [R1]- [R3].

[R1] Tarnai, T. (2001). Kinematic bifurcation. In *Deployable Structures* (pp. 143-169). Vienna: Springer Vienna.

[R2] Dai J S, Rees Jones J. Mobility in metamorphic mechanisms of foldable/erectable kinds. *Journal of Mechanical Design*, 1999, 121(3): 375-382.

[R3] Zhang, L., & Dai, J. S. (2009, June). An overview of the development on reconfiguration of metamorphic mechanisms. In *2009 ASME/IFToMM International Conference on Reconfigurable Mechanisms and Robots* (pp. 8-12). IEEE.

Reply: We agree that the term “Bifurcation” is used in multiple fields including mathematics, mechanics, stability analysis, kinematics, etc. and can have different meanings under different context. The dictionary definition of “bifurcation” is “the division of something into two branches or parts”, which describes the phenomenon observed in our work well. However, as the reviewer has pointed out, the term “kinematic bifurcation” does have a more specific meaning under the context of mechanisms analysis, where there is usually no participation of “locking devices”. Thus, to avoid any potential confusion or misunderstanding, we replace the term “kinematic bifurcation” with “multi-path folding motion” and the related text is revised appropriately. These changes are marked blue.

Finally, there are quite a few unproven conclusions. Examples are not limited to those listed below:

(1) "we show that uniform thickness of MUTO systems enables them to carry large loads." There is no direct evidence that the load-carrying capacity comes from uniformly thick-panel, it seems more likely that it comes from the locking devices.

Reply: Thank you for raising this constructive criticism. From a structural engineering point of view, having uniform thickness is ideal for load carrying because the flow of forces is aligned. While this concept is generally well-known and agreed upon in the structural engineering community, we agree that it is unfair to make such a claim without substantiating it. Thus, to address this comment, the following comparative experiment is added to the manuscript to prove the above claim.

In this experiment, we compared the stiffness and the strength of a hinged connection with uniform thickness with a similar connection with uneven thickness. The connectors are built with the same materials and have the same minimum thickness. Our results show that uniform thickness connection can achieve an ultimate load that is twice as large as the connection with uneven thickness (2.7 kN of tensile force versus 1.3kN). Moreover, the connection with uniform thickness has an initial stiffness of 5.1 GPa, while the connection with uneven thickness has an initial stiffness of only 1.1 GPa. We believe these experimental data provide concrete evidence to prove that uniform thickness is necessary for good load-carrying performance.

The following sentence is added to the main text to address this comment:

“Supplementary Text Section S9 shows an experiment to demonstrate superior load-carrying performance of hinged origami connections with uniform thickness. In this comparative experiment, the uniformly thick hinge is five times stiffer and has twice the ultimate capacity as compared to a similar system with uneven thickness.”

The above figure is added as a new supplementary Fig S22, and new text is included in supplementary section S9 to discuss the experiment.

(2) "we have observed two trends: first, origami structures that can fold cannot carry large loads [25, 26], and second, origami structures that can carry large loads cannot fold [27, 28, 29]. Moreover, common origami patterns for civil applications tend to have only one kinematic path [30, 25], so they are not well suited for creating adaptable systems that offer multiple configurations." The authors have missed some of the origami research that are foldable and capable of bearing utilizing self-locking or bifurcation ([R4]-[R9]).

[R4] Ye, H., Liu, Q., Cheng, J., Li, H., Jian, B., Wang, R. ... & Ge, Q. (2023). Multimaterial 3D printed self-locking thick-panel origami metamaterials. Nature Communications, 14(1), 1607.

[R5] Jamalimehr A, Mirzajanzadeh M, Akbarzadeh A, et al. Rigidly flat-foldable class of lockable origami-inspired metamaterials with topological stiff states[J]. Nature communications, 2022, 13(1): 1816.

[R6] Zhao, Z., Kuang, X., Wu, J., Zhang, Q., Paulino, G. H., Qi, H. J., & Fang, D. (2018). 3D printing of complex origami assemblages for reconfigurable structures. *Soft Matter*, 14(39), 8051-8059.

[R7] Zhai, Z., Wang, Y., & Jiang, H. (2018). Origami-inspired, on-demand deployable and collapsible mechanical metamaterials with tunable stiffness. *Proceedings of the National Academy of Sciences*, 115(9), 2032-2037.

[R8] Tomita, S., Shimanuki, K., Nishigaki, H., Oyama, S., Sasagawa, T., Murai, D., & Umemoto, K. (2023). Origami-inspired metamaterials with switchable energy absorption based on bifurcated motions of a Tachi-Miura polyhedron. *Materials & Design*, 225, 111497.

[R9] Yasuda H, Gopalarethinam B, Kunimine T, et al. Origami - based cellular structures with in situ transition between collapsible and load - bearing configurations[J]. *Advanced Engineering Materials*, 2019, 21(12): 1900562.

Reply: Thank you for this comment and the suggested references. We want to point out that these suggested references all focus on metamaterial type systems, with the maximum dimension smaller than 1 meter. None of these references provide a solution to develop deployable meter-scale civil structures, such as a beam that can span 4 meters or more.

The reviewer's comment above quotes an incomplete sentence from our manuscript which is highly misleading. The full sentence as included in our manuscript clearly indicates the distinction that we are focused on large-scale origami for civil engineering or architectural applications. The full sentence is included here.

“Despite the tremendous progress in multiple fields, when it comes to large-scale origami for civil engineering or architectural applications, we have observed two trends: first, origami structures that can fold cannot carry large loads [25, 26], and second, origami structures that can carry large loads cannot fold [27, 28, 29]...”

Nonetheless, in response to this comment, we have added the following sentence to page 3 of the manuscript, to highlight the newly mentioned references and their relevance to this work. Many of these papers were already included in the updated literature review during the last revision cycle.

“More specifically, origami and kirigami systems can utilize self-locking to form load-bearing materials [24, 25, 26] or use MDOF kinematics to achieve different configurations [27, 28, 10, 11]. However, these systems are targeting metamaterial applications or small-scale mechanisms with prototypes smaller than one meter.”

(3) The author's reply "Our work ...explores modularity of MUTOs to achieve high adaptability, harnesses kinematic bifurcations for system reconfiguration, and uses experimental tests to demonstrate the high strength and stiffness of these structures. These characteristics were not previously demonstrated in the field of structural origami." The previous works in [R10-R14] with kirigami, or modular origami techniques are essentially folding something using the reconfigurability of multi-DOFs systems or kinematic bifurcation of a single DOF system.

[R10] Li Y, Zhang Q, Hong Y, et al. 3D transformable modular Kirigami based programmable metamaterials. *Advanced Functional Materials*, 2021, 31(43): 2105641.

[R11] Li Y, Yin J. Metamorphosis of three-dimensional kirigami-inspired reconfigurable and reprogrammable architected matter. *Materials Today Physics*, 2021, 21: 100511.

[R12] Yamaguchi K, Yasuda H, Tsujikawa K, et al. Graph-theoretic estimation of reconfigurability in origami-based metamaterials. *Materials & Design*, 2022, 213: 110343.

[R13] Liu W, Jiang H, Chen Y. 3D programmable metamaterials based on reconfigurable mechanism modules[J]. *Advanced Functional Materials*, 2022, 32(9): 2109865.

[R14] Wang C, Li J, Zhang D. Motion singularity analysis of the thick-panel kirigami[J]. *Mechanism and Machine Theory*, 2023, 180: 105162.

Reply: Thank you for this comment and the suggested references. Although the suggested references do show origami systems with MDOF kinematics and multiple configurations, these systems are not targeting large-scale civil structures. The mentioned references are primarily targeting metamaterial systems, with maximum dimension smaller than 1 meter.

In response to this comment, we have added the following sentence to page 3 of the manuscript. To highlight the mentioned references and their relevance to this work. Many of these papers were already included in the updated literature review during the last revision cycle.

“More specifically, origami and kirigami systems can utilize self-locking to form load-bearing materials [24, 25, 26] or use MDOF kinematics to achieve different configurations [27, 28, 10, 11]. However, these systems are targeting metamaterial applications or small-scale mechanisms with prototypes smaller than one meter.”